



# Modelling the impact of trapped lee waves on offshore wind farm power output

Sarah J. Ollier[1], Simon J. Watson[2]

[1]CREST – Loughborough University, Holywell Park, Loughborough LE11 3TU, UK
[2]Wind Energy Section, Faculty of Aerospace Engineering, Delft University of Technology, Kluyverweg 1, 2629 HS Delft, Netherlands.

*Correspondence to*: Sarah J Ollier (s.ollier@lboro.ac.uk)

**Abstract.** Mesoscale meteorological phenomena, including Atmospheric Gravity Waves, or Trapped Lee Waves (TLWs) can result from flow over topography or coastal transition in the presence of stable atmospheric stratification, particularly with

strong capping inversions. Satellite images show that topographically forced TLWs frequently occur around near-coastal offshore wind farms. Yet current understanding of how they interact with individual turbines and whole farm energy output is limited. This parametric study investigates the potential impact of TLWs on a UK near-coastal offshore wind farm, Westermost Rough (WMR) resulting from westerly – south-westerly flow over topography in the Southeast of England.

Computational fluid dynamics (CFD) modelling (using ANSYS-CFX) of TLW situations based on real atmospheric

conditions at WMR was used to better understand turbine level and whole wind farm performance in this parametric study based on real inflow conditions. These simulations indicated that TLWs have the potential to significantly alter the windspeeds experienced by and the resultant power output of individual turbines and the whole wind farm. The location of the wind farm in the TLW wave cycle was an important factor in determining the magnitude of TLW impacts, given the expected wavelength of the TLW. Where the TLW trough was coincident with the wind farm, the turbine windspeeds and power outputs were more

substantially reduced compared with when the TLW peak was coincident with the location of the wind farm. These reductions were mediated by turbine windspeeds and wake losses being superimposed on the TLW. However, the same initial flow conditions interacting with topography under different atmospheric stability settings produce differing near wind farm flow. Factors influencing the flow within the wind farm under the different stability conditions include differing: hill and coastal transition recovery, windfarm blockage effects and wake recovery. Determining how much of the differences in windspeed

and power output in the wind farm resulted from the TLW is an area for future development.

## 1   Introduction

Atmospheric Gravity Waves (AGWs) often result from displacement of flow by topographical obstacles in neutral or stable surface atmospheric conditions with a strong temperature inversion above the atmospheric boundary layer. They also form via jet stream turbulence, weather fronts, cold air outbreaks, thunderstorms, tornadoes, hurricanes, polar lows and other unknown

sources (Gossard and Hooke, 1975; Rasmussen and Aakjær, 1992; Romanova and Yakushkin, 1995; Chunchuzov et al., 2000;





Nappo, 2012). The flow displaced by these conditions oscillates to create waves which modulate the local wind speed. AGWs are frequent in the offshore environment and influence marine atmospheric boundary layer wind fields over large areas of the ocean, (e.g. Thomson et al., 1992; Vachon et al., 1994).

Strong stable capping temperature inversions aloft, often induced by changes in temperature at the coastal transition, provide a 'lid' to trap the waves created by topographical obstacles, resulting in horizontally propagating AGWs, known as Trapped Lee Waves (TLWs). In the last 12 years, AGW propagation instigated by windfarms themselves has been investigated (e.g. Smith, 2010; Allaerts and Meyers, 2017a, 2019, 2017b; Allaerts et al., 2018; Lanzilao and Meyers, 2020) and the impact of TLWs on onshore windfarms has recently been investigated (Xia et al., 2021; Draxl et al., 2021; Wilczak et al., 2019). However, to our knowledge, no one has investigated the influence of pre-existing TLWs on individual turbines or whole wind

farms offshore and computational fluid dynamics (CFD) investigations of TLW-wind farm investigations have not been published. Considering their influence on offshore wind speeds, TLWs are likely to impact offshore wind power production. Thus, this research investigates the influence of TLWs on offshore wind farm power output using Reynolds Averaged Navier Stokes (RANS) CFD simulations. Influences on the flow under differing stability conditions are summarised in Fig. 1.

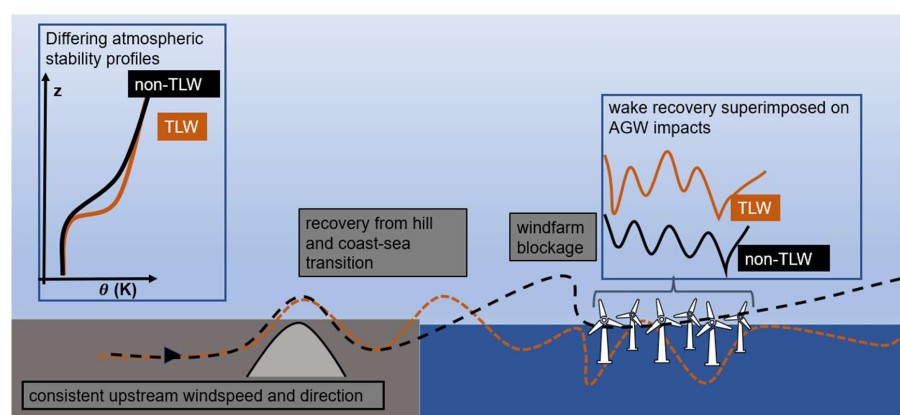


**Figure 1. Interaction of consistent windspeed and direction with different stability conditions upstream of a topographical obstacle and an offshore wind farm. The dashed lines show the evolution of flow aligned with a single column of wind turbines. This flow evolves under different stability conditions, a strong capping inversion for the TLW (orange) and a conventionally neutral boundary layer (CNBL, black) without TLWs (non-TLW). Insets show stability profiles (left) and wind farm wake recovery superimposed on**

**the background flow for a single column of turbines aligned with the prevailing wind direction.**

We use a theoretical offshore wind farm downstream of a topographical obstacle to simulate the impact of TLWs on the wind power output. Although the set-up is theoretical, the layout used is based in the operational offshore wind farm at Westermost Rough (WMR) off the East Yorkshire coast. This theoretical windfarm referred to as WMR throughout this paper. The following section covers identification of TLW conditions at WMR (section 2.1), sections 2.2-2.7 describe the modelling

methodology in ANSYS CFX for TLWs at WMR. Section 3 presents and discusses the modelled impact of TLWs on the



turbines and windfarm, the implications of TLWs on WMR are summarised in section 4 and suggested future investigations are included in section 5.

## 2    Methodology

### 2.1 TLW identification

SAR data from Sentinel 1a/b, pre-processed for 10m wind (DTU Wind Energy, 2021), were used to detect TLW events at Westermost Rough offshore windfarm (WMR, Fig. 2).

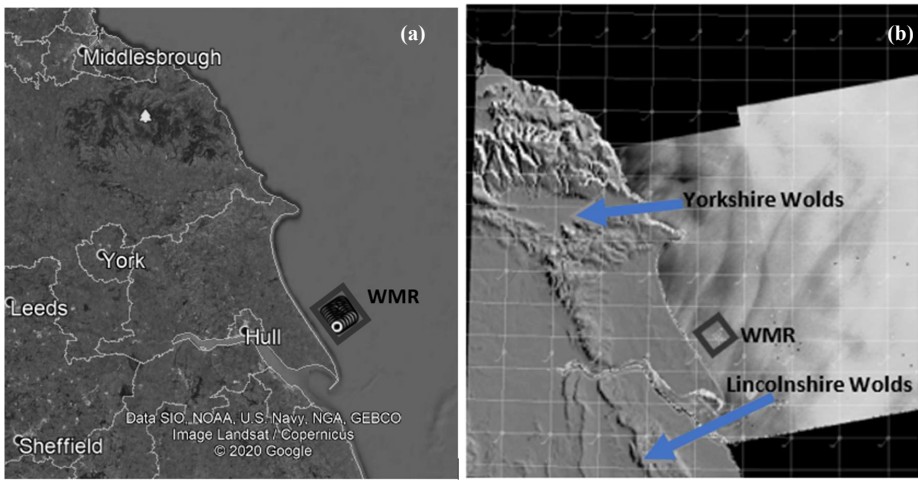

**Figure 2. a): WMR location off the Holderness coast of North East England, WMR shown by grey polygon (Data SIO, NOAA, U.S. Navy, NGA, GEBCO, Image Landsat/Copernicus ©2020 @2021 Google).b) SAR image of WMR (DTU Wind Energy, 2016), raised**
**topography shown by darker shades of grey, Location of WMR shown by grey polygon.**

Sentinel 1a/b passed over WMR every 1-3 days in 2016 – 2017 around 06:15 or 17:45 UTC. The SAR images were visually inspected for TLWs in a similar manner to other studies (e.g. Li et al., 2013b; Li, 2004; Xu et al., 2016). TLW classification of images was based on the appearance of a repeating linear pattern of fluctuating windspeeds, perpendicular to the prevailing wind direction at the location of WMR. A potential temperature vertical profile proxy for the site was taken from 97 vertical

levels of ERA5 reanalysis data (ERA5, 2020) for the lowest 5 km of the atmosphere. The existence of a strong temperature inversion in ERA5 was used to confirm the likelihood of TLW formation. A TLW event at WMR was selected to provide the boundary conditions for CFD simulations and a CNBL event with a weak inversion, not strong enough to produce TLW, was selected as a control (Section 2.6).



### 2.2 Domain and Topography

For all RANS simulations, ANSYS-CFX 18.0 was used with ANSYS Windmodeller as a front end to set up the simulations. The topography includes a simplified representation of a steep near coastal ridge as in (Ollier et al., 2018), based on a two-dimensional hill profile. The hill dimensions are based on a 'Witch of Agnesi' profile (Fig. 3).

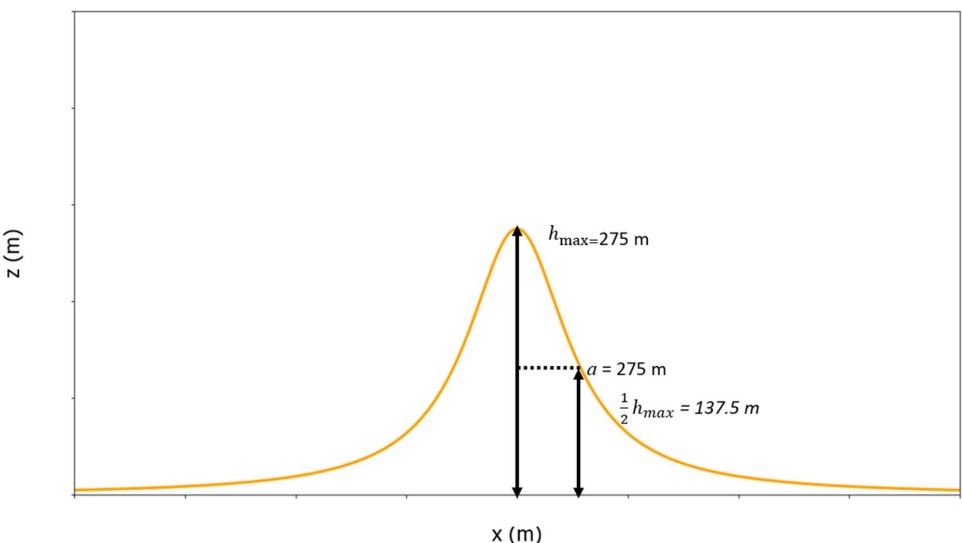

**Figure 3. Witch of Agnesi shaped profile used in coastal ridge simulations (not to scale).**

The hill height $h(x)$ depends on the maximum hill height $h_{max}$, (chosen as 275 m) and half-width at half-height $a$, (also chosen as 275 m) as a function of horizontal distance $x$ from the centre of the hill (Eq 1). This results in a very steep hill, with a slope $\sim 65$ % ($\sim33°$). N.B. Flow separation is expected at slopes $\geq 30$ %.

$$h(x) = \frac{h_{max} \cdot a^2}{x^2 + a^2}$$

This is a very simplified hill compared to the actual topography upstream of WMR. Due to the complexity of the real terrain upstream of WMR, the simplified hill model does not attempt to capture the terrain features other than that of a simple hill,

which is the same distance from the wind farm as WMR is from the coast with the aim of inducing TLWs.

This two-dimensional hill is elongated to form a ridge aligned perpendicular to the incoming westerly (270°) wind. The hilltop is 11 km from the inlet (Fig. 4a). There is flat coastal terrain at an elevation of 10 m above sea level (asl) upstream of the ridge, with a constant roughness length ($z_0$) of 0.03 m. The sea with constant roughness length ($z_0 = 0.0002$ m) is located downstream of the coastline as shown in (Fig. 4). All domains have an upper and outlet Rayleigh damping region (section 2.3).



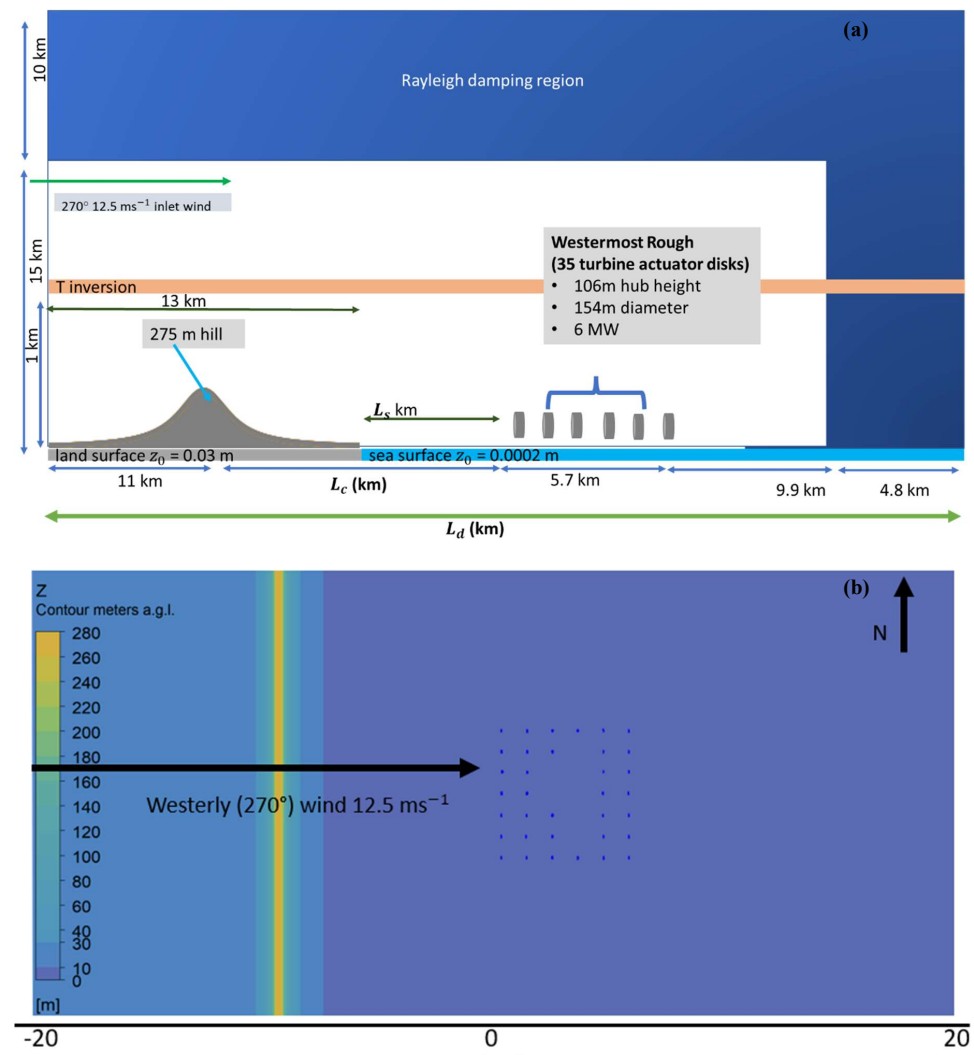


Figure 4. a) Diagram of the WM simulation domain (not to scale) for the coastal hill cases at WMR. For the 41.4 km domain $L_s, L_c$ and $L_d$ are 8 km, 10 km and 41.4 km, respectively. For the 46 km domain $L_s, L_c$ and $L_d$ are 12.5 km, 14.6 km and 46 km, respectively (not to scale). b) view from WM domain for WMR coastal hill case, see legend for heights (m). Each yellow dot represents a single
turbine location (axes is not to scale).



To capture TLW peak interactions with WMR, a 41.4 km long domain was used (Fig. 4). This cuboid domain is 41.4 km long, 20 km wide and 25 km high (Fig. 4). These dimensions allow for insertion of topography and downstream actuator discs representing WMR. The flow reaches equilibrium before and after these obstacles before reaching the domain boundaries. Many studies use spanwise infinite wind farms, but as (Allaerts and Meyers, 2017a) notes, this exaggerates the blockage effect and the excitation of wind farm produced TLWs, likely overestimating their strength. The present study allows for flow transport around the model of a finite wind farm. The height of the incompressible flow domain (25 km) was chosen to avoid non-physical numerical reflections of the gravity waves (section 2.3).

An extended domain of 46 km was used to assess the impact of TLWs hitting the wind farm at the trough of the TLW rather than the peak (Fig. 4). This domain follows the same layout as the 41.4 km domain but the distance between the hill and the wind farm was extended by 4.6km (Fig. 4), approximately half the TLW wavelength modelled in the 41.4 km domain simulations.

**2.3 Wave Damping**

Rayleigh damping at the domain boundaries was introduced to prevent unphysical wave reflections. Rayleigh damping, which absorbs waves before they can be reflected at domain boundaries, was first introduced in early, two-dimensional mountain wave models (Klemp and Lilly, 1978; Durran and Klemp, 1983) through the use of a simple damping term depending on the perturbation of a variable from its equilibrium value. In the current work, the damping coefficient was split into two components, $\tau_z(z), \tau_x(x)$ (Eqs. 2-4) as a function of the x and z coordinates respectively, and damping ($\xi_w$) was added only to the right-hand side of the $z$-momentum equation using Eq. 5. This was done to provide damping layers at the top and outlet of the domain:

for $H_{max} \geq z \geq 0$:

$$\tau_z(z) = \tau_{0z} \exp\left(-0.5 \left(3.5 \; \frac{z - H_{max}}{z_\tau}\right)^2\right) \qquad \textbf{2}$$

for $x < 0$:

$$\tau_x(x) = 0 \qquad \textbf{3}$$

for $x \geq 0$:

$$\tau_x(x) = \tau_{0x} \exp\left(-0.5 \left(3.5 \; \frac{x - r_\tau}{D_\tau}\right)^2\right) \qquad \textbf{4}$$

for $H_{max} \geq z \geq 0$ and $x \geq 0$:

$$\xi_w = (\tau_x(x) + \tau_z(z)) \, w \qquad \textbf{5}$$

The constants $\tau_{0x}$ and $\tau_{0z}$ (units of s$^{-1}$) are set equal to 1 kg m$^{-3}$ s$^{-1}$/$\rho_0$, where $\rho_0$ is air density at sea level (1.23 kg m$^{-3}$) resulting in a maximum damping at the domain top and outlet of 0.8 s$^{-1}$. $x$ is the horizontal location in the domain (m) (x = -20700, 20700 for the 41.4 km domain, Fig. 4b), $r_\tau$ is the distance from the centre of the domain where damping is implemented (20.7



km for the 41.4 km domain and 23 km for the 46 km domain), $D_\tau$ is the characteristic horizontal length for which damping is applied (4.8 km, all domains e.g. Fig. 4a, $H_{max}$ is the maximum domain height and $z_\tau$ the characteristic vertical depth for which damping is applied. This latter value was chosen to correspond to 40% of the domain height, i.e., 10 km, in line with a recent LES study (Allaerts and Meyers, 2017a). In the same study, low wave reflection was also reported when there was space for at least one vertical wavelength, $\lambda_z$, beneath the damping layer (Allaerts and Meyers, 2017a), based on an earlier linear

model (Klemp and Lilly, 1978), where $\lambda_z$ is defined as:

$$\lambda_z = \frac{2\pi U}{N} \qquad \textbf{6}$$

where $U$ is the bulk windspeed and $N$ is the freestream Brunt-Väisälä frequency (Eq. 7).

$$N = \sqrt{\frac{g}{\theta}\frac{\partial \theta}{\partial z}} \qquad \textbf{7}$$

where $g$ is gravitational acceleration in (ms$^{-2}$), $\theta$ is potential temperature (K) and $\frac{\partial \theta}{\partial z}$ is the free atmosphere lapse rate.

In the current work the domain height is 3.2 $\lambda_z$.


Whilst damping layer strength, depth, location and how the damping layers are implemented varies between studies, so do the atmospheric conditions (windspeed and inversion strength), domain dimensions, grid resolution, topography, wind farms sizes and layouts. Thus, it is not possible to directly compare the methodologies and deduce the optimal conditions to transfer to other studies. Recent studies (Ollier et al., 2018; Jia et al., 2019) used RANS to model TLWs, but do not detail their damping

methodology. The literature discussed here uses LES configurations rather than RANS, some of which include a precursor domain (Gadde and Stevens, 2019; Wu and Porté-Agel, 2017), unlike the current work. Domain top Rayleigh damping strength ranged from 0.0001s$^{-1}$ – 0.016s$^{-1}$ in (Allaerts and Meyers, 2017a; Gadde and Stevens, 2019; Haupt et al., 2019; Hills and Durran, 2012) with a range of upper level damping thicknesses (1–16 km). A three-dimensional mountain ridge was included in (Hills and Durran, 2012), and no wind farm was included. A damping layer of 16 km in the vertical, from z = 20 km to the

domain extent (z = 36 km) was used in a very large domain (1200 km x 1200 km x 36 km), no inflow or outflow damping was included. TLW reflection was observed with stronger domain top damping in (Hills and Durran, 2012). The maximum damping in this layer was 0.005 s$^{-1}$, gradually increasing from 0 s$^{-1}$ outside the layer.

Damping near domain inlet/outlet boundaries is not consistent in all studies; some use an outflow damping, some do not; and some have inflow and outflow damping. In two-dimensional models containing a simple hill and no turbines (Haupt et al.,

2019), inflow and outflow damping layers were not important for the solution in very long domains (200 km). However, upper-level damping was essential for the same domains, with optimal damping of 0.005 s$^{-1}$. Further, outflow damping reduced spurious upstream waves in shorter domains, but did not eliminate them. With shorter upstream distance, with damping at the inflow, outflow and upper level, an LES model showed reasonable agreement with an analytical solution (Haupt et al., 2019).





Unfortunately, the domain dimensions were not included to contextualise these findings. Quasi-stationary topographic TLWs
were modelled in a relatively shallow LES domain (22 km x 19 km x ~3 km) with 3 km high complex mountain terrain (Li et
al., 2013a). Interestingly, no problems with wave reflection were reported, and damping was not discussed.

Notable wind farm LES studies with wind farm induced TLWs include (Allaerts and Meyers, 2017a; Maas and Raasch,
2022; Wu and Porté-Agel, 2017; Smith, 2010). The study in (Allaerts and Meyers, 2017a) used 10km upper-level damping
(0.0001 s⁻¹) and 4.8km outflow damping (0.03 s⁻¹) applied with a gradually increasing cosine profile within a 38.4 x 4.8 x 25
km domain. The domain contained a spanwise infinite wind farm (180 regularly spaced turbines) over a sea surface of constant
$z_0$, 0.0002m.

In the absence of consistent guidance in the literature regarding the optimal set up of Rayleigh damping layers, the best
configuration, used for all the domains in this chapter, was based on modification of the default Rayleigh damping in ANSYS
Windmodeller. The damping strength was unchanged but the location and thickness of the damping layers were modified. This
was determined by trial and error. Whilst the damping layer strength used in this research is higher than in previous studies,
Durran and Klemp (1983) found that the depth of the damping layer more important and damping strength did not strongly
influence the solution.

**2.4 Boundary conditions**

At the inlet (Western plane), Dirichlet boundary conditions (i.e., prescribed profiles) were applied for the velocity vector, the
potential temperature θ and the turbulence quantities (turbulence kinetic energy $k$ and turbulence dissipation rate ε). For the
pressure, a zero-gradient condition is applied. The inlet profiles for the relevant variables were set up as follows: below the
boundary layer height, $h_{BL}$, the velocity profile follows a log profile, while above it, the profile is set to the velocity value at
the top of the boundary layer, $V_G$. With the flow directed along the x axis, velocity profiles (Eq. 8-9) were used for the velocity
components $(V_x, V_y, V_z)$:

$$V_x(z) = min\left(\frac{u_*}{\kappa}\ln\left(\frac{z}{z_{0,us}}\right), V_G\right)$$

$$V_y = V_z = 0$$

Where $z_{0,us}$ is the surface roughness upstream. The von Kármán constant, $\kappa$ is set to a value of 0.41. The roughness length
$z_0$ is used to set the profile by calculating the friction velocity ($u_*$, Eq. 10). The boundary layer height is calculated from the
empirical relationship in Eq. 10 (Garratt, 1994).

$$h_{BL} = 0.25\frac{u_*}{f}$$




With f is the Coriolis parameter ($1.2 \times 10^{-4}$ s$^{-1}$). The inlet profiles for the turbulence kinetic energy and dissipation rate are defined in Eq. s 11-13:

$$k(z) = \ max\left[\frac{u_*^2}{\sqrt{C_\mu}}\left(1 - \frac{z}{h_{BL}}\right)^{1.68}, 10^{-4} \ m^2/s^2\right]$$ **11**

$$\varepsilon(z) = \ max\left[\frac{u_*^3}{\kappa z} 1.03 \ F_{cor} \ exp\left(-2.8\left(\frac{z}{h_{BL}}\right)^2\right), 10^{-4} \ m^2/s^3\right]$$ **12**

$$F_{cor} = \ \left[1 + \frac{0.015}{z^{0.9}} max\left(0, ln\frac{z}{z_0}\right)\right]$$ **13**

where $F_{cor}$ is a roughness dependent correction factor. The profiles for the turbulence quantities in Eq. s 11-13 are approximate fits to numerical results obtained for a one-dimensional simulation of a developing boundary layer over the sea, after 24 hours of physical time (Montavon et al., 2012).

At the outlet (Eastern plane) and at the top of the domain, an entrainment opening boundary condition is used which applies: zero-gradient condition on the velocity, zero-gradient on the potential temperature and turbulence quantities when the flow is 185 locally out of the domain. If the flow is entering the domain at those locations, the model then applies the same prescribed profiles as those used for the inflow. A Dirichlet boundary condition for the pressure, where the prescribed pressure profile is calculated to satisfy the hydrostatic balance associated with the potential temperature profile applied at the inflow[1]. At the sides of the domain (Northern and Southern planes), symmetry conditions are used for all variables. At the ground, no-slip boundary conditions are used for the velocity, using wall functions to characterise the momentum fluxes as a function of the 190 local roughness length and friction velocity $u_*$ (ms$^{-1}$, Eq. 14) (ANSYS, 2017):

$$u_* = C_\mu^{1/4} k^{1/2}$$ **14**

where $C_\mu$ is the turbulence model constant (0.09).

For neutral surface layer simulations (section 2.6), adiabatic (i.e., zero heat flux) conditions are used for the potential temperature and for the turbulence kinetic energy. Where surface stability is included, diabatic (heat flux) conditions are used for potential temperature and for the turbulence kinetic energy.

---

[1] When no flow prevails in the domain, the momentum conservation equation in the vertical is simplified to $\frac{\partial p}{\partial z} = g\rho\frac{1}{\theta_0}(\theta - \theta_0)$. The pressure profile used at the outflow is calculated by integrating this relationship from the ground to the top of the domain, using the prescribed profile for $\theta_{in}$ at the inflow, and the reference potential temperature $\theta_0$. When using a pressure profile not satisfying the hydrostatic balance, the model generates flow acceleration or slow-down that can destabilise the solution. .



All simulations use a 270° 12.5 ms⁻¹ reference wind speed at the turbine hub height (106 m). The closure for the turbulence dissipation rate at the ground is provided by ε, (Eq. 12). Atmospheric stability conditions are detailed in section 2.5. The Coriolis force has been shown to deflect wakes in wind farms and wake deflection is more pronounced in stable boundary layers (e.g. Gadde and Stevens, 2019). However, to isolate the effects of stability and the Coriolis effect, the Coriolis force is 'switched off' for all simulations.

This model assumes isotropic turbulent viscosity where the ratio of Reynolds stress and rate of deformation is equal in all directions. Whilst the $k$-$\varepsilon$ RANS model is less accurate in the near wake region (e.g. Argyle, 2014), for the whole farm simulations in this research the far wake is more important. The $k$-$\varepsilon$ turbulence model uses modified $C_\mu$ (0.03, Eq. 14) (Montavon et al., 2011) for all simulations as it performed best in preliminary trials, increasing the eddy viscosity in turbine wakes and reducing numerical noise in TLW simulations.


### 2.5 Atmospheric conditions

For the simulations including atmospheric stability , the freestream potential temperature gradient, was set to 3.3 x 10⁻³ K km⁻¹ in line with the International Standard Atmosphere (ISO 2533:1975). The potential temperature profile is set as follows:

for $z < z_{inv} - d$:

$$\theta_{in} = \theta_1 \qquad\qquad 15$$

for $z_{inv} - d < z < z_{inv}$:

$$\theta_{in} = \theta_1 \ + (\partial\theta/\partial z)_{inv}[z - (z_{inv} - d)] \qquad\qquad 16$$

$$(\partial\theta/\partial z)_{inv} = (\theta_0 + (\partial\theta/\partial z)_0 . z_{inv} - \theta_1)/d \qquad\qquad 17$$

for $z > z_{inv}$:

$$\theta_{in} = \theta_0 \ + (\partial\theta/\partial z)_0 z \qquad\qquad 18$$


where $z$ is height, $z_{inv}$ is height of the top of the inversion layer, $d = z_{inv} - dz_{inv}$ is the inversion layer depth and $dz_{inv}$ the inversion base. $(\partial\theta/\partial z)_{inv}$ is the lapse rate for the temperature inversion and $(\partial\theta/\partial z)_0$ is the free atmosphere lapse rate (Montavon, 2017; Ollier et al., 2018), $\theta_{in}$ is the potential temperature at the inflow, $\theta_0$ , the reference potential temperature and $\theta_1$ , potential temperature for $z$ at the inflow.

Neutral atmospheric stability was used as a control for the 41.4 km and 46 km domains. In these cases, the atmospheric stability conditions were neutral throughout, with a constant potential temperature of 288 K (Ollier et al., 2018), (purple dots, Fig. 6). These simulations were given the short code '0N' (Table 1, Table 2). TLW simulations included a capping inversion with lapse rate $(\partial\theta/\partial z)_{inv}$ = 7.6 K km⁻¹ and a stable surface layer (short code '7S', e.g. r7Sh-WMR, Table 1, Table 2;, the

temperature profile was based on the atmospheric conditions during a TLW event at WMR (Fig. 5a, Fig. 6Fig. 5). In the ERA5

data, there was a temperature inversion around 1km – 2.5km with $(\partial\theta/\partial z)_{inv}$= 7.8 K km$^{-1}$ (blue crosses, Fig. 6). This ERA5

profile also had a stable surface layer with an approximately -2 K surface offset, increasing to near neutral at z ~300 m. A -2K

surface temperature offset was applied in the 7S simulations gradually increasing to neutral at z ~30 m at the inlet. As the

profile develops in the domain this vertical distance increases to 300 m, comparable to the ERA5 stable surface depth. The

temperature inversion was introduced using Eq. s 15-18, (Fig. 5, (Montavon, 2017; Ollier et al., 2018)) with the following

parameters: $z_{inv}$= 2.5km, $z_{inv} - d$ = 1km, $(\partial\theta/\partial z)_{inv}$ = 7.6 K km$^{-1}$ (Fig. 5, Fig. 6) at the surface. This is the basis for proxy

atmospheric conditions for TLW formation, conditioned on the potential temperature profile, wind direction and windspeed at

a reference height 106 m (turbine hub height). To assess the impact of the stable surface layer, the same temperature inversion

with a neutral surface layer was included (green dash line Fig. 6, r7Nh-WMR, Table 1, Table 2).

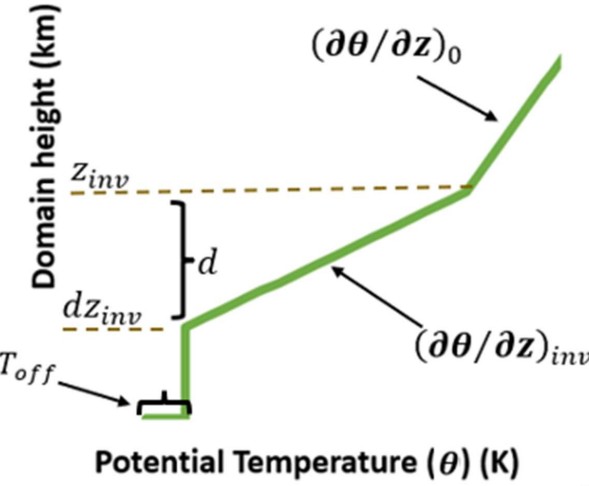

**Figure 5. Potential temperature schematic used in the simulations with stability where z is height, $z_{inv}$ is height of the top of the inversion layer, d is the distance between the top and bottom of the inversion layer. $(\partial\theta/\partial z)_{inv}$ is the lapse rate for the temperature inversion and $(\partial\theta/\partial z)_0$ is the lapse rate above the inversion and $T_{off}$ is the surface temperature offset.**

For a control simulation based on real atmospheric conditions at WMR, the weak CNBL simplified profile (gold dashes, Fig. 6) was used. This is based on the weak CNBL event identified in SAR (Fig. 7b). The ERA5 data (blue crosses, Fig. 6)

was taken from the same location as the capping inversion TLW case. The inversion base is at 0.6 km, with a 3.3 K km$^{-1}$ lapse rate. As $(\partial\theta/\partial z_{inv})$ = 3.3 K km$^{-1}$ is the same as the freestream potential temperature gradient $(\partial\theta/\partial z_0)$, there is not an upper limit to the inversion. These simulations were given the short code '3N' (Table 1).



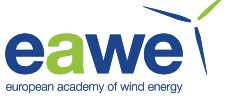
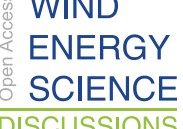

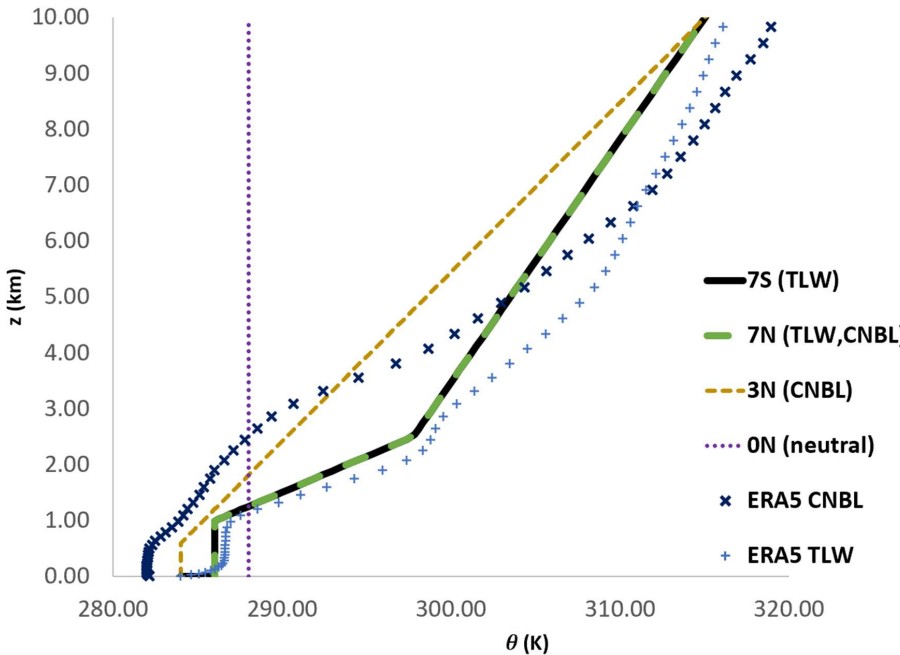

**Figure 6. Stability profiles from ERA5 non-TLW (CNBL) (blue cross) and TLW events (blue diamond) and WM inflow conditions approximating to the same events. Where 7S is $(\partial\theta/\partial z)_{inv}$ = 7.6 K km$^{-1}$ with stable surface layer, 7N is $(\partial\theta/\partial z)_{inv}$ = 7.6 K km$^{-1}$ with neutral surface layer. 3N is $(\partial\theta/\partial z)_{inv}$ = 3.3 K km$^{-1}$ with neutral surface layer and 0N is neutral throughout. Short codes summarised in Table 1.**




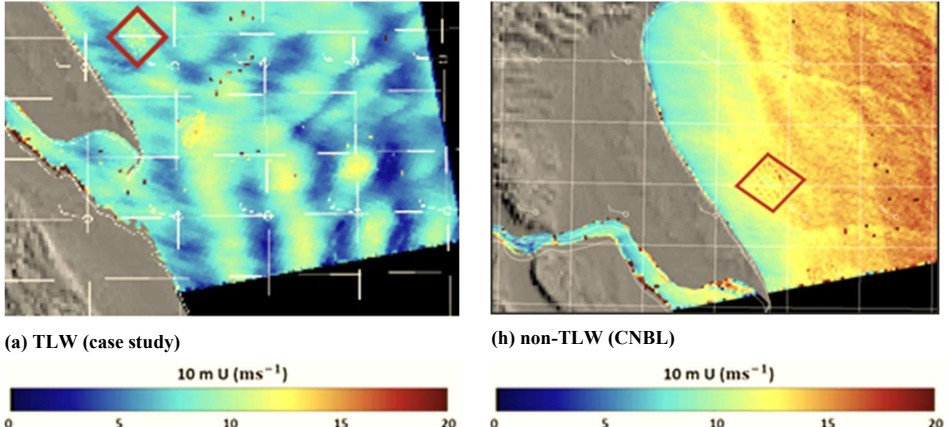

**(a) TLW (case study)**          **(h) non-TLW (CNBL)**

**Figure 7. Examples of TLWs and non-TLW events detected in SAR images at WMR. Red polygon shows location of WMR. Legend shows 10 m windspeeds (ms$^{-1}$). Images adapted from (DTU Wind Energy, 2021, 2016).**

**Table 1. Short codes for simulations**

| Short code | definition |
|---|---|
| domain length | |
| r | Regular domain (41.4 km) |
| x | Extended domain (46 km) |
| Capping inversion | $\partial\theta/\partial z_{inv}$ (K km$^{-1}$), $dz_{inv}$ (km), $z_{inv}$ (km) |
| 7 | 7.6, 2.5, 1.5 |
| 3 | 3.3, 25 (domain extent), 0.6 |
| 0 | No inversion, neutral conditions throughout |
| Surface stability | |
| S | Stable surface layer |
| N | Neutral surface layer (CNBL) |
| Topography | |
| h | Coastal hill |
| Wind farm | |
| WMR | WMR wind farm |
| NWF | No wind farm |






**Table 2. Overview of simulations**

| Simulation | Inlet U ($ms^{-1}$) | Stability $\partial\theta/\partial z_{inv}$ ($K\,km^{-1}$) | $z_{inv}$ (km) | $d$ (km) | Surface stability | $dz_{inv}$ (km) | turbines (number, layout) | topography | dimensions x,y,z (km) |
|---|---|---|---|---|---|---|---|---|---|
| r7Sh-WMR (TLW peak) | 12.5 | 7.6 | 2.5 | 1 | stable -2 K | 1.5 | 35 WMR layout | coastal hill, ocean | 41.4 x 20 x 25 |
| r7Sh-NWF | 12.5 | 7.6 | 2.5 | 1 | stable -2 K | 1.5 | - | coastal hill, ocean | 41.4 x 20 x 25 |
| r7Nh-WMR | 12.5 | 7.6 | 2.5 | 1 | neutral | 1.5 | 35 WMR layout | coastal hill, ocean | 41.4 x 20 x 25 |
| r3Nh-WMR | 12.5 | 3.3 | 25 | 3 | neutral | 0.6 | 35 WMR layout | coastal hill, ocean | 41.4 x 20 x 25 |
| r0Nh-WMR | 12.5 | - | - | - | neutral | - | 35 WMR layout | coastal hill, ocean | 41.4 x 20 x 25 |
| x7Sh-NWF | 12.5 | 7.6 | 2.5 | 1 | stable -2 K | 1.5 | - | coastal hill, ocean | 46 x 20 x 25 |
| x7Sh-WMR (TLW trough) | 12.5 | 7.6 | 2.5 | 1 | stable -2 K | 1.5 | 35 WMR layout | coastal hill, ocean | 46 x 20 x 25 |





### 2.6 Turbine set up

The WMR layout and spacing was used for all simulations (Fig. 8). The WMR layout was rotated by 33° to align with the 270° inlet wind in the domain (Fig. 4, Fig. 8). This alignment is equivalent to south westerly winds reaching WMR at turbine row A.

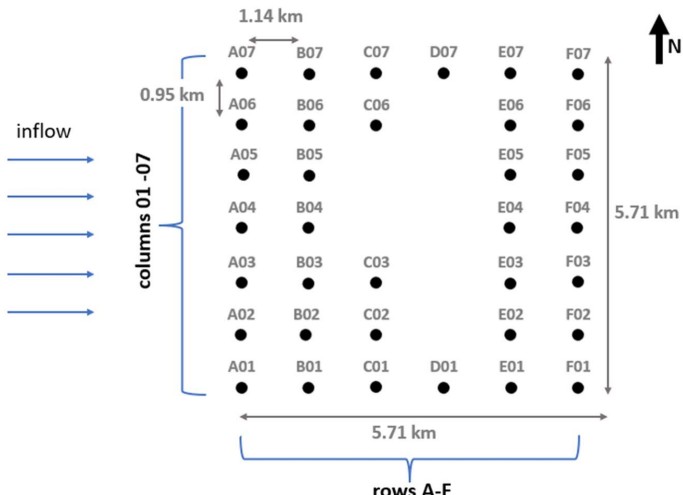

**Figure 8. WMR layout for WM domain, using the same spacing as WMR but rotated 30° to align with the 270 degree wind direction**
**in the domain (the equivalent of SW flow reaching WMR). Rows and columns labelled as referred to in the text.**

Turbines were modelled as actuator discs (ADs) whose thrust is conditioned on the upstream wind speed and the thrust curve is modified to be a function of disc (as opposed to freestream) wind speed. The actual hub heights (106 m), spacing (0.95 within row, 1.14 km between rows) and rotor diameters (154 m) of WMR turbines were used in this model (Fig. 8), with AD

thickness ~38.5 m. All turbines were set to be operational during the simulations and to yaw to the local flow direction (Ollier et al., 2018). A 6MW, 154m diameter turbine theoretical power curve was used with thrust data for a Siemens 3.6 MW direct drive wind turbine (SWT-3.6-107) (Appendix A). For individual turbines, local turbulence intensity is determined by Eq. 19. The freestream turbulence intensity offshore was 0.07 for all simulations.

$$TI = \frac{\sqrt{\frac{2}{3}k}}{U_{hub}}$$

Where $k$ is turbulent kinetic energy, $U_{hub}$ is the windspeed at the turbine hub.




Turbine $U_{us}$ is obtained by using Actuator Disc theory to convert $U_{hub}$ (Eq. 2019) to $U_{us}$.

$$U_{hub} = U_\infty(1 - a_i)$$

where:

$$a_i = \frac{1}{2}(1 - \sqrt{1 - C_T(U_\infty)})$$

Where $U_{hub}$ is the windspeed at the turbine rotor, $C_T$ is the thrust coefficient, $U_\infty$ is the freestream windspeed and $a_i$ is the axial induction factor.

Turbine meshing was set up as in Ollier et al., (2018). The background horizontal resolution (outside of the rotor regions) for the model domain is 60 m (Appendix B). 150 vertical levels were used, and the first cell above ground is 2 m thick with a geometric mesh expansion factor of 1.15 for the levels above. For the simulations containing turbines, the Windmodeller built-in mesh adaption algorithm was selected for a finer mesh around the turbines. This includes approximately 15 cells across a 154 m diameter rotor, corresponding to approximately 10.3 m per cell (Appendix B). Mesh refinement restriction was applied

around the turbine actuator discs, to avoid an unnecessarily fine mesh away from the turbine locations, thus reducing numerical noise and computational cost.

The simulations used in the current work are summarised in Table 2.

**3     Results and discussion**

**3.1 Trapped Lee Waves**

For all simulations, the inlet windspeed is 12.5 ms$^{-1}$ at a reference height of 106 m. However, the windspeeds just upstream of the wind farm vary due to flow evolution throughout the domain with differing atmospheric stability conditions interacting

with terrain and turbines. For comparison of windfarm inflow conditions, near upstream windspeed ($U_N$, Fig. 9) refers to windspeeds at a point 300 m upstream of the bottom row of WMR before the blockage effect occurs (x = 0 for 41.4 km domain, Fig. 9). The wind farm blockage effect varies under the different stability conditions described in sections 3.1-3.3. The labels in Fig. 9 illustrate the different influences on $U_N$.




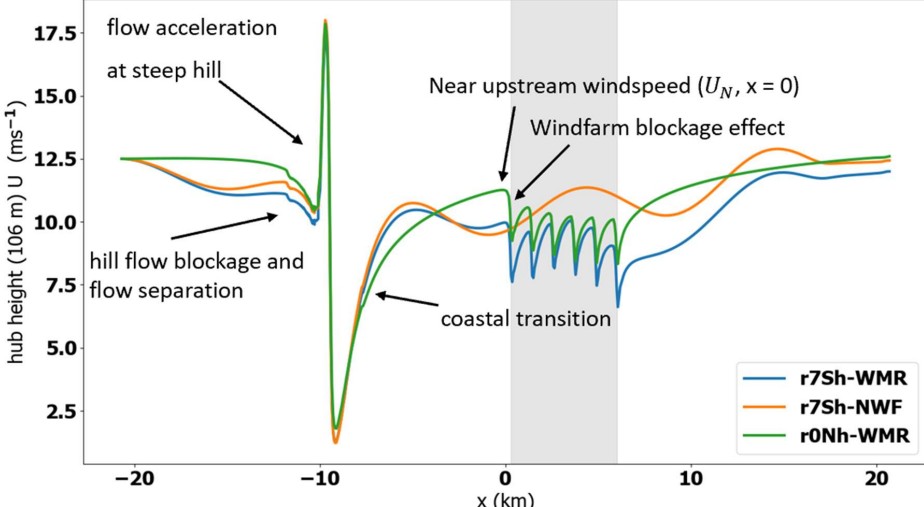

**Figure 9. Wind farm and stability impacts on the flow, Values are at 106 m above the surface showing the TLW peak case at WMR (r7Sh-WMR, orange line), the TLW case without a wind farm (7Sh-NWF, blue line), neutral control with WMR (r0Nh-WMR, green line). Grey shaded region shows the x location of WMR wind farm.**

Due to the variation in the values of $U_N$ for the different simulations, direct comparisons between the simulation $U_{us}$, power, inflow angle and TI are complicated by different turbine thrust values. Absolute values are not compared in the current work, but the relative flow and power properties will still be influenced by differences in location on the turbine thrust and power curves at the given windspeeds. Despite this limitation, these results demonstrate topographical TLW impacts on flow patterns and consequent power outputs across WMR. Some of the influences on both $U_N$ and turbine windspeed and power which are difficult to decouple are discussed in sections 3.1-3.3 including: recovery from the topographical blockage effect, presence of topographic TLWs, TLW phase, capping inversion and surface stability impacts, coastal transition flow adjustment impacts, presence, or absence of upstream TLWs, and windfarm flow blockage effect.




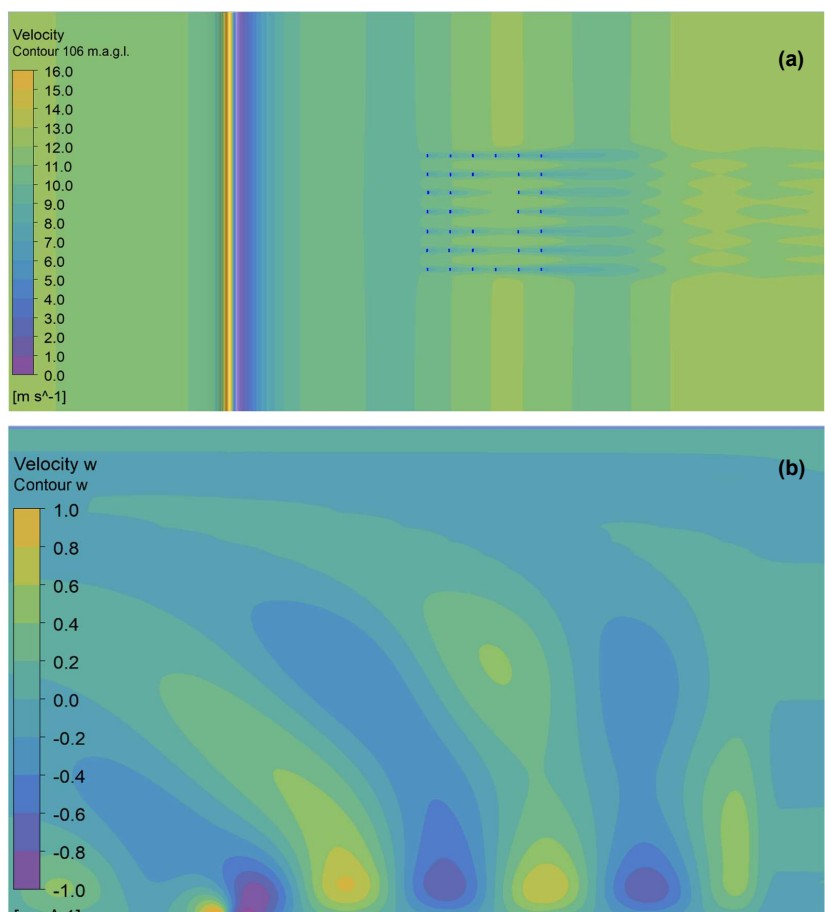

**Figure 10. a) view from above: Horizontal velocity at 106m above the surface throughout the TLW (r7Sh-WMR) simulation domain. b) Side view: vertical velocity throughout the simulation domain the TLW (r7Sh-WMR) simulation aligned with the column 01 of WMR.**


For the coastal hill simulations at WMR where $(\partial\theta/\partial z)_{inv}$ = 7.6 K km$^{-1}$ (r7Sh-WMR, r7Sh-NWF, Table 3.2), TLWs are observed downstream of the hill and persist throughout the domain to the outflow in both the horizontal velocity (Fig. 10a) and vertical velocity fields (Fig. 10b). Notably, there is a TLW peak upstream of the hill in Fig. 10b; TLW peaks also occurred in mathematical models of wind-farm induced TLWs where the Froude number ($Fr$, Eq. 22) was less than 1 (Smith, 2010;

Lanzilao and Meyers, 2021).



$$Fr = \left[\frac{U}{NH}\right] \qquad 22$$

Where $U$ is mean windspeed (ms⁻¹), $H$ is the obstacle height (m).

In the TLW cases (r7Sh-WMR, r7Nh-WMR) $Fr \sim 0.04$, so the upstream wave behaviour is consistent with findings in previous studies (Smith, 2010; Lanzilao and Meyers, 2021). However, it is unclear at this stage whether the upstream peak is

an artefact of imperfect wave damping and upstream domain length. The upstream peak is, however, considered far enough upstream of the windfarm to have negligible impact on the solution at WMR. The flow decelerates rapidly on approach to the steep hill ridge (slope ~33°, Fig. 3), with acceleration and flow separation at the peak and lee side (Fig. 10, Fig. 11). The flow separation is quite severe owing to the steepness of the hill. In the 41.4 km domain, the flow is still recovering from this deceleration upon approach to WMR. The TLW is superimposed on the recovering flow and has a gradually increasing

windspeed (Fig. 10, Fig. 11a). The TLW characteristics are similar for the simulations with (r7Sh-WMR) and without (r7Sh-NWF) WMR wind farm, but the interaction with WMR results in overall lower windspeeds than when it is absent (Fig. 11a). TLW peak windspeeds are 11.4 ms⁻¹ (r7Sh-NWF) and 10 ms⁻¹ (r7Sh-WMR), with a mean difference of 0.94 ms⁻¹ throughout the domain. The mean windspeed for the TLW case (r7Sh-WMR) is lower throughout the domain than for the neutral situation (r0Nh-WMR). In part this is due to the faster recovery from the hill wake in the neutral case (Fig. 11a). For reference,

windspeeds for the neutral case (r0Nh-WMR) are included in Fig. 11. With a stable surface and capping inversion present (r7Sh-WMR) the flow recovery from the steep hill is slower so the TLW begins with a much lower windspeed than the neutral case. This discrepancy in $U_N$ makes absolute comparison unclear. Further, it is not possible to fully decouple the impact of wind farm blockage effect under a strong $\partial\theta/\partial z_{inv}$ compared to neutral, where the blockage appears less (Fig. 11a). Two full TLW cycles are apparent in Fig. 10 and Fig. 11.





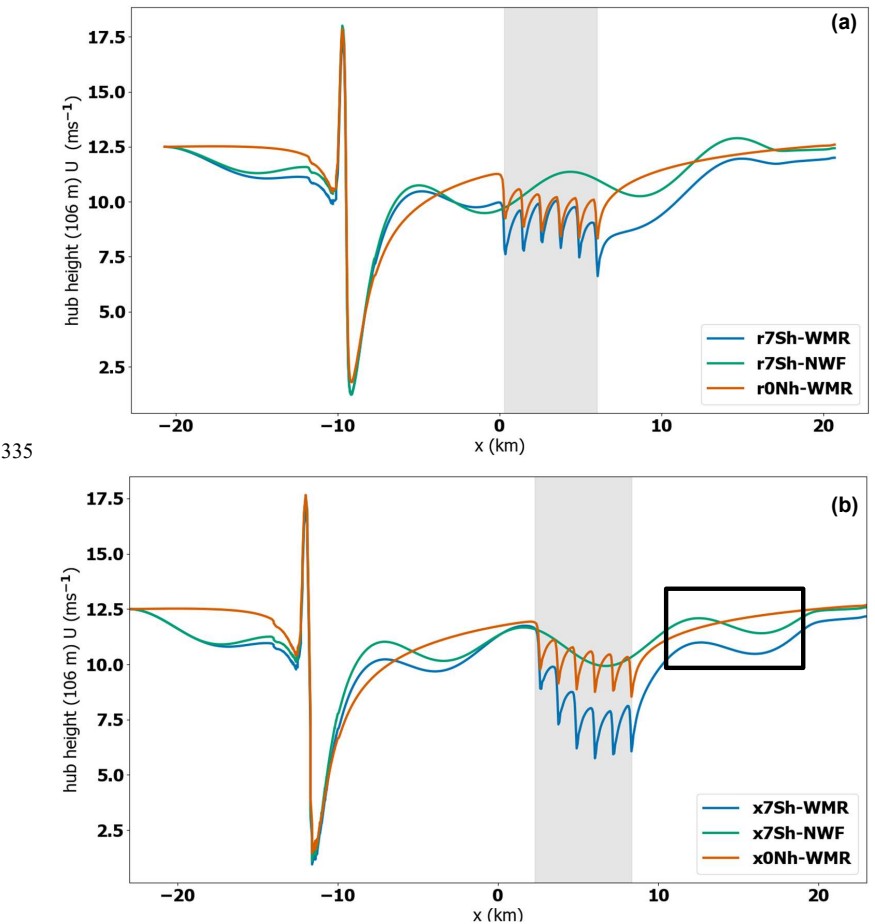

**Figure 11. Wind speed at 106 m above the surface U (ms⁻¹). a) for TLW peak with WMR (r7Sh-WMR, blue line) TLW peak without wind farm (r7Sh-NWF, orange line) and neutral case with WMR (r0Nh-WMR, green line) in the regular domain. b) 46 km domain for the TLW trough under the same conditions (x7Sh-WMR, x7Sh-NWR) and neutral case (x0Nh-WMR). Grey shaded region shows the x location of WMR wind farm. Black box highlights area of amplitude difference between TLW simulations.**

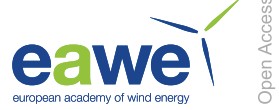
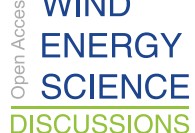

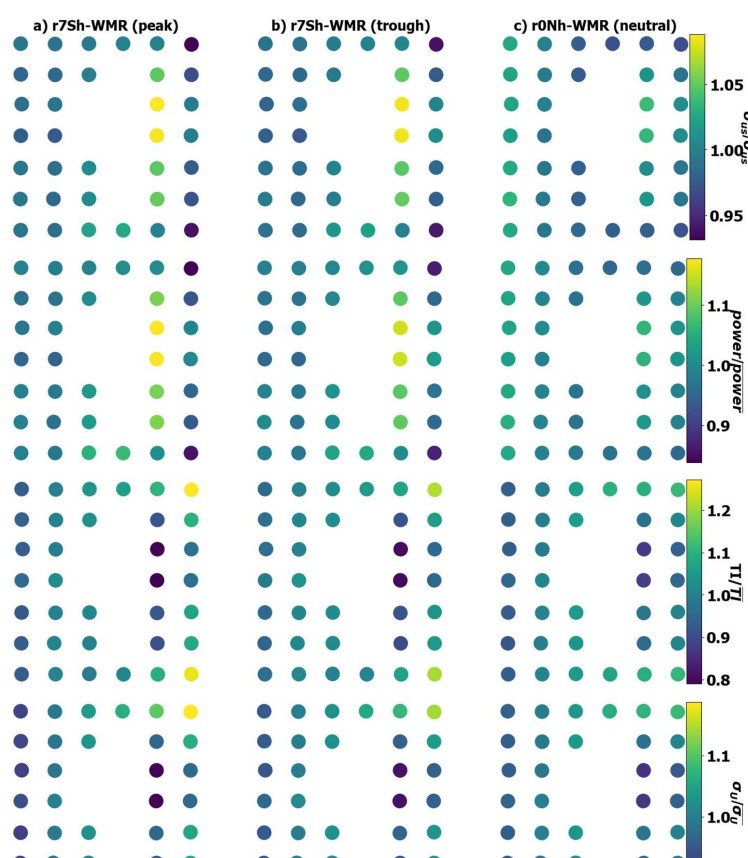

**Figure 12.** Values normalised to the mean for all turbines for $U_{us}$ (ms$^{-1}$), power output, turbulence ($\sigma_U$) and TI and inflow angle (°) for the coastal hill domain. TLW peak (column a, r7Sh-WMR), TLW trough (column b, r7Nh-WMR) and neutral conditions (column c, r0nh-WMR)




Whilst both the neutral and TLW cases show recovery in windspeeds after the central gap in WMR (Fig. 12), the increase in windspeeds is much higher in the TLW case. The TLW has a maximum 10% increase in $U_{us}$ (perpendicular velocity of the air cylinder upstream of the actuator disc) between turbines in row B and E either side of the gap in the TLW situation (r7Sh-WMR, Fig. 12). For the neutral situation (r0Nh-WMR) there is only 4% increase for the same turbine locations (Table 3). The

greater recovery is explained by the increases in windspeeds due to the TLW countering wake losses. However, the central gap in WMR makes the TLW effect less clear. Furthermore, the variability in $U_{us}$ throughout WMR is considerably higher for the TLW case (r7Sh-WMR) than the neutral case, with the range of windspeeds experienced by the turbines over double that of the neutral simulation (Fig. 12, Table 3). The TLW range of $U_{us}$ and power output across the farm are 2.1 and 2.4 times the neutral case respectively (Table 3). The power difference is greater due to the non-linear nature of the power and thrust curves.

There are also coincident greater increases in turbulence and local TI (Eq. 19) at the turbines when the TLW is present (Fig. 12), as the trend is the same for both parameters this is attributed to more variable vertical velocity and shear in the TLW situation which are influenced by the coupled impacts of the TLW and the capping inversion (Appendix C).

Column 01 of WMR, where there is no gap between turbines, is less affected by adjacent columns under the 270° wind. This column shows the clearest TLW signature (Fig. 12a). Throughout this column, the range of $U_{us}$ values is 0.9 ms$^{-1}$ for the

TLW case compared to 0.6 ms$^{-1}$ in the neutral case. For the same locations, mean $U_{us}$ is 1.2 ms$^{-1}$ less for the TLW case than neutral. The range in power output down column 01 for the TLW is over double the neutral case (1000 kW, 472 kW, respectively). This is due in part to differences in windspeed position on the thrust and power curves between the simulations exaggerating the windspeed differences.




**Table 3: Simulation descriptive statistics for all WMR turbines for $U_{us}$, power, TI, inflow angle and shear exponent factor ($\alpha$).**

| | | r0Nh-WMR | r7Sh-WMR | r7Nh-WMR | x7Sh-WMR | r3Nh-WMR | x0Nh-WMR |
|---|---|---|---|---|---|---|---|
| $U_{us}$(ms⁻¹) | mean | 11.4 | 10.7 | 10.9 | 10.2 | 10.2 | 11.7 |
| | max | 11.8 | 11.6 | 11.9 | 11.9 | 10.7 | 12.3 |
| | min | 11.1 | 9.9 | 10.3 | 9.1 | 9.7 | 11.2 |
| | range | 0.8 | 1.7 | 1.6 | 2.8 | 0.9 | 1.1 |
| | std | 0.2 | 0.4 | 0.3 | 0.9 | 0.3 | 0.3 |
| **Power (kW)** | mean | 5400 | 4700 | 5000 | 4200 | 4200 | 5600 |
| | max | 5700 | 5500 | 5700 | 5700 | 4700 | 5900 |
| | min | 5100 | 3900 | 4300 | 3000 | 3700 | 5200 |
| | range | 600 | 1600 | 1400 | 2700 | 1000 | 700 |
| | std | 200 | 300 | 300 | 900 | 300 | 200 |
| | total | 188200 | 164200 | 174100 | 146600 | 147100 | 196000 |
| **Turbine TI** | mean | 0.18 | 0.18 | 0.18 | 0.24 | 0.19 | 0.16 |
| | max | 0.20 | 0.23 | 0.22 | 0.18 | 0.23 | 0.19 |
| | min | 0.16 | 0.14 | 0.15 | 0.25 | 0.16 | 0.13 |
| | range | 0.04 | 0.09 | 0.08 | 0.11 | 0.07 | 0.05 |
| | std | 0.01 | 0.02 | 0.02 | 0.14 | 0.02 | 0.02 |
| $\sigma_u$ | mean | 2.0 | 1.9 | 2.0 | 1.8 | 2.0 | 1.8 |
| | max | 2.2 | 2.3 | 2.3 | 2.3 | 2.3 | 2.1 |
| | min | 1.8 | 1.6 | 1.7 | 1.3 | 1.7 | 1.6 |
| | range | 0.4 | 0.6 | 0.5 | 1.0 | 0.5 | 0.5 |
| | std | 0.1 | 0.1 | 0.1 | 0.3 | 0.2 | 0.1 |
| **Inflow angle (°)** | mean | 0.53 | 0.74 | 0.77 | 1.03 | 0.83 | 0.52 |
| | max | 0.63 | 1.50 | 1.31 | 1.62 | 0.93 | 0.58 |
| | min | 0.45 | 0.20 | 0.44 | 0.64 | 0.73 | 0.43 |
| | range | 0.17 | 1.30 | 0.87 | 0.98 | 0.20 | 0.14 |
| | std | 0.04 | 0.34 | 0.25 | 0.29 | 0.06 | 0.03 |
| $\alpha$ | mean | 0.13 | 0.18 | 0.18 | 0.21 | 0.17 | 0.12 |
| | max | 0.18 | 0.25 | 0.26 | 0.37 | 0.24 | 0.16 |
| | min | 0.10 | 0.12 | 0.11 | 0.09 | 0.12 | 0.08 |
| | range | 0.08 | 0.14 | 0.15 | 0.28 | 0.12 | 0.09 |
| | std | 0.02 | 0.04 | 0.04 | 0.07 | 0.03 | 0.02 |

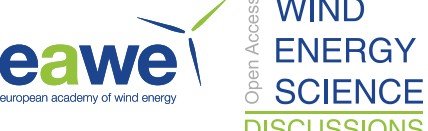

### 3.2 Location of WMR in TLW wave cycle

Whilst the wave behaviour is similar in the 41.4 km domain (r7Sh-WMR) and the extended 46 km (x7Sh-WMR) domain, the
flow characteristics at WMR are notably different depending on where the TLW hits the wind farm (Fig. 11). Turbine
windspeeds and wakes in column 01 of WMR increase and decrease in phase with the TLW. In both cases, the TLW shape is
clearly superimposed on the turbine windspeeds, despite the wind farm blockage effect and the fluctuation within the wind
farm due to wake losses (Fig. 11). With stable surface conditions, wake recovery is slower, yet the TLW reduces the impact
of the wake losses when windspeeds increase towards the peak of the wave, counteracting some of the surface stability
influence (r7Sh-WMR, Fig. 11a). Wake losses are, however, amplified towards the TLW trough (x7Sh-WMR, Fig. 11b).

When the TLW is reaching its trough (x7Sh-WMR, Fig. 11b), TLW reduction in windspeeds compounds the reduction in
wind speed due to the wake losses so the windspeeds are dramatically reduced. These windspeed reductions are much more
pronounced than the reductions after the TLW peak in r7Sh-WMR. This is explained by differences in $nU_{us}$ between
simulations. The turbines in the trough case experience the trough windspeeds at a steeper location on the thrust curve leading
to deeper wake losses. At the wind farm level (Fig. 12), mean $U_{us}$ is reduced relative to the neutral situation in both the peak
and trough situations, as even in the peak case the first turbine row (row A) is in the recovery from an upstream TLW trough.
Here with the same far upstream conditions, but the TLW hitting the wind farm at a different part of the wave cycle, the range
in windspeeds is 1.7 times the range for trough compared to peak case (Table 3.3); this difference is of the same order as the
difference between the peak TLW and the neutral case. The difference in $U_N$ is 1 ms$^{-1}$, so the windspeed range difference is
explained mainly by the wave positioning, exaggerated by operating at a different point on the thrust curve, rather than the $U_N$
alone.

In the extended domain, the neutral simulation has a slighter higher $U_N$ compared to the TLW case (11.9 ms$^{-1}$ and 11.7 ms$^{-1}$,
respectively, Fig. 11b). This is due to the longer distance between the hill and WMR allowing for further windspeed recovery
from the steep hill. Consequently, there is also a much smaller difference in $U_{us}$ (mean 0.4 ms$^{-1}$) in turbine column 01, between
the TLW (x7Sh-WMR) and neutral case (x0Nh-WMR), than in the regular domain. Therefore, it is possible to directly compare
the windspeeds between the two cases. The large range in $U_{us}$ throughout WMR for the TLW situation (2.8 ms$^{-1}$, x7Sh-WMR,
Table 3.3) is mainly accounted for by the atmospheric conditions rather than differences in initial $U_{us}$, with a mean difference
of 1.4 ms$^{-1}$ at WMR between the two cases. The large range in $U_{us}$ could be attributed to increased wake losses due to the
stable surface conditions and the strong capping inversion aloft rather than the TLW itself.

At the wind farm level (Fig. 12) the TLW signature is most clearly seen down column 01 of WMR in both cases, which is
less affected by the gap within the centre of the wind farm. The wave pattern is subtly apparent throughout all the turbine rows
with $U_{us}$ and subsequent power output rising and falling in phase with the TLW cycle (Fig. 12). TI varies more in both the
peak (TI range 0.09) and the trough cases (TI range 0.11) with both ranges over double the neutral case (TI range 0.04, Table
3). The changes in TI have a similar distribution to the changes in turbulence, suggesting that the range of turbulence is a result
vertical velocity changes in the TLW rather than windspeed differences. As the vertical velocity is more variable during TLW



flows, so are the inflow angles compared to neutral conditions (see Table 3 and Fig. 12). Notably, the TI, shear and turbulence are higher in the peak case for turbines F01 and F07, this is discussed in Appendix C.

Regardless of where in the wave cycle the TLW interacts with WMR, it recovers and the wave train persists after interaction with WMR with a slight reduction in windspeed compared to the no wind farm scenario (r7Sh-NWF, x7Sh-NWF, Fig. 11).

The TLW appears to flatten at the domain outlet, but this is due the outlet wave damping. This suggests the same topographical TLW may cause deviations from predicted power output for multiple wind farms downwind of the same hill or coastline. This is similarly discussed for onshore wind farms in (Draxl et al., 2021). However, due to the domain length here, it is not possible to see how far the TLW wave-trains persist and how much windspeeds recover downstream.

These results demonstrate the TLW impact on the flow, $U_{us}$, power, TI, and inflow angles throughout WMR, but to

understand the impact it is essential to determine which part of the wave cycle the wind farm is in when experiencing quasi-stationary gravity waves. The impacts of the TLW will fluctuate in severity across the wind farm with TLW phase. As location in the TLW phase has such a pronounced impact, this suggests the wind farm dimensions and turbine spacing will also be important as they will affect how much of the wind farm is within the TLW. Similarly, the wavelength and amplitude will determine what proportion of a given wind farm is in the different TLW phases and how severe the windspeed changes

are.

WMR interaction with the TLW appears to have negligible impact on wavelength; the distance between the TLW peak in the WMR centre (grey shaded region, Fig. 11a) and the first peak after the WMR the wavelength is ~10.7 km for both the WMR and NWF situation at 600 m.a.s.l. away from the turbine rotors. This is also the case for the TLW trough situations. This is comparable to the wavelength predicted from the upper layer Scorer parameter ($l^2$, Eq. 23-24, ~12 km).

$$\lambda = \frac{2\pi}{l(z)} \qquad\qquad \textbf{23}$$

$$l^2(z) = \frac{N^2}{U^2} - \left(\frac{\partial^2 U}{\partial z^2}\right) / U \qquad\qquad \textbf{24}$$

Where $\lambda$ is wavelength, $N = N(z)$, $U = U(z)$ is the vertical profile of the horizontal wind.

There are apparent reductions in TLW amplitude where WMR is present compared to the NWF simulations. However, these differences are superimposed on flow recovery and wind farm blockage effects. The difference is most clearly observed in the black box in Fig. 11b where the peak-trough amplitude windspeed difference is 0.7 and 0.5 ms⁻¹ for x7Sh-NWF and x7Sh-WMR, respectively. Amplitude reduction is also observed upstream of WMR in Fig. 11a,b. As the TLW persists with

reduced amplitude after interaction with WMR, this suggests that a TLW event affecting multiple farms may have less impact on windspeed and power fluctuations if there is another windfarm upstream.

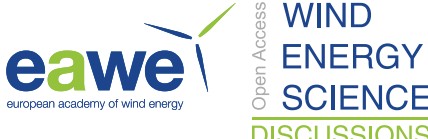

### 3.3 Surface layer stability impacts

Whilst the section above discusses the impact of a temperature inversion with stable surface layer (r7Sh-WMR), this section investigates whether the stable surface layer has a strong effect on the variation of windspeeds across the wind farm. For r7Sh-WMR the stable surface layer has less impact than might be expected as the profile becomes neutralised as it evolves through the domain (Fig. 13b). Using Monin-Obukhov Similarity Theory (Eq. 25) and taking $\frac{z}{L}$ across blade tip heights (29-183 m) between the inlet and hill (line 1, Fig. 13), gives $\frac{z}{L}$ = 1.09 and $L$ = 96.9 m suggesting the flow is very stable.

$$\frac{z}{L} = Ri_G \qquad\qquad 25$$

where $z$ is height, $L$ is the Obukhov length and $Ri_G$ is the gradient Richardson number.

Yet the stability profile has a relatively subtle temperature offset once the inlet profile has adjusted within the domain and interacted with the topography to a relatively small temperature offset (< 1 K) and shallower surface layer (lines 2-11, Fig. 13b). Changes in velocity profile after the topography may be attributed to TLW trough flow effects on shear and associated
turbulence as described in Vosper et al., (2018). To obtain a temperature profile with strong stability at WMR the Windmodeller inlet surface temperature offset would need to be increased to counteract the neutralisation in the domain.





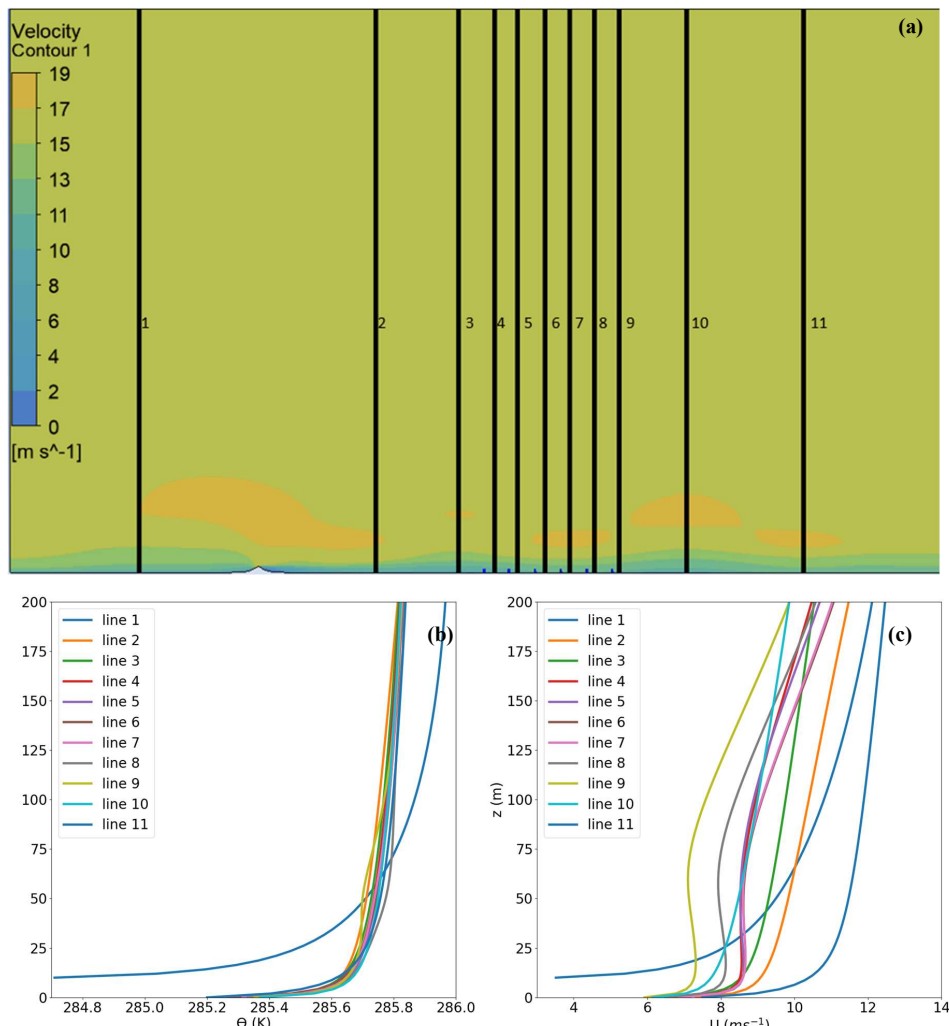

Figure 13. a) vertical slice through the domain in line with column 01 of WMR turbines showing vertical velocity for the AGW case (r7Sh-WMR). Yellow lines represent are lines 1 –11 as labelled in all 3 plots. Below: WM potential temperature (b) and velocity (c) profile for lines 1-11.


As shown in Fig. 14, the flow throughout the domain is similar for both the inversion case with the neutral surface layer (r7Nh-WMR) and stable surface layer (r7Sh-WMR). With the stable surface layer (r7Sh-WMR), mean $U_{us}$ and power are reduced

with increased wake losses slightly increasing the windspeed reduction effect of the AGW. This leads to negligible increases in variation in $U_{us}$ with a range of $U_{us}$ which is 0.07 ms$^{-1}$ greater for r7Sh-WMR than for the neutral surface layer (r7Nh-
WMR) with resulting power output variation range of 1592 kW (for r7Sh-WMR) and 1435 kW (for r7Nh-WMR) (Table 3, Fig. 14, Fig. 15).

The influence of the AGW dominates with a slight reduction in the range of values for all variables for r7Nh-WMR compared to the stable surface layer case (r7Sh-WMR, Table 3). Fig. 15 compares the whole wind farm for the stable surface and neutral surface cases (power, TI, $U_{us}$) with neutral conditions for the regular domain. As the surface stability temperature
offset reduces substantially after interaction with the topography and sea surface (lines 2-11, Fig. 13), the stable layer is relatively shallow with the surface lapse rate increasing to near neutral conditions around rotor height. Thus, the differences between r7Sh-WMR and r7Nh-WMR are relatively subtle. In these situations, the impact of the capping inversion appears much more important than the surface stability. Yet, much larger differences between the stable and neutral surface layer simulations would be expected with a stronger and deeper stable layer at the surface which would increase wake losses further.

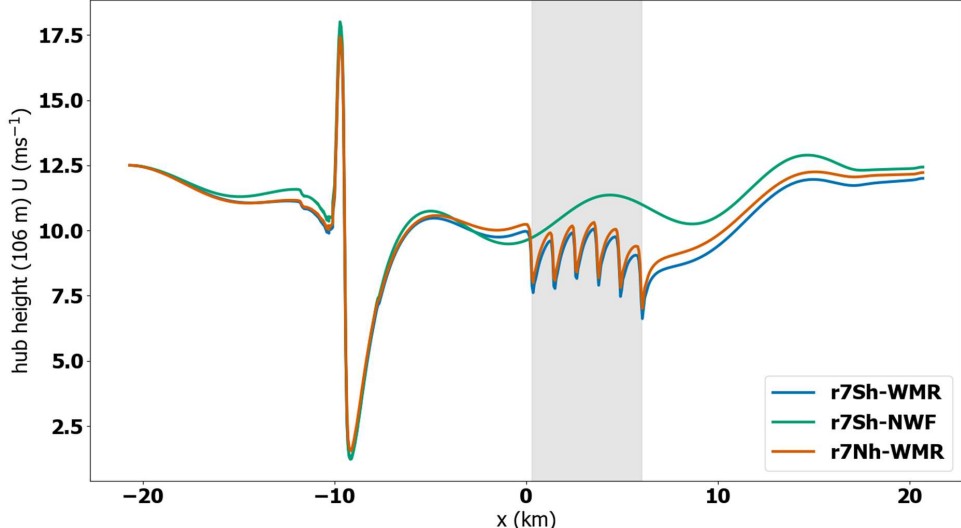


**Figure 14. 106 m above the surface isolines of U aligned with column 01 of WMR for capping inversions with (r7Sh-WMR, orange line) and without (r7Nh-WMR, blue line) stable surface conditions and stable surface conditions without WMR (r7Sh-NWF, green line).**





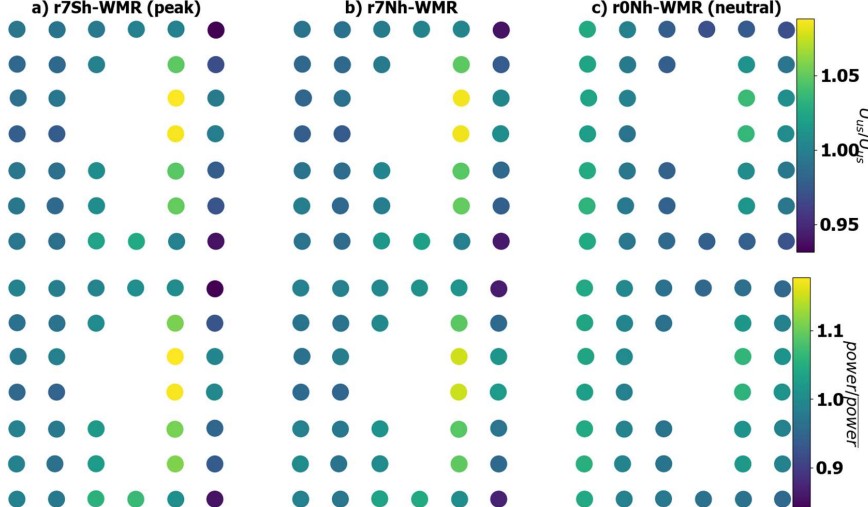

**Figure 15. View from above WMR, normalised to the mean value for WMR for $U_{us}$ and power output (kW) for the AGW peak (column 01, r7Sh-WMR), AGW peak CNBL (column b, r7Nh-WMR), and neutral (column c) r0Nh-WMR) cases.**


### 3.4 TLW compared to CNBL conditions at WMR

The CNBL case (r3Nh-WMR) is a more realistic atmospheric situation than purely neutral conditions (r0Nh-WMR). In Fig. 16a $U_N$ for r3Nh-WMR is substantially higher than for the TLW peak situation (r7Sh-WMR, 10.9 ms$^{-1}$, 9.6ms$^{-1}$, respectively

at x=0). However, the gradual decline in windspeeds due to wake losses reduces $U_{us}$ throughout WMR to less than those for the TLW situation (r7Sh-WMR, b), where the losses are countered by the peak of the TLW. The reduced windspeed in the CNBL case (r3Nh-WMR) results in a lower position on the power curve, resulting in a reduced power output across WMR compared to the TLW peak and neutral cases (Fig. 16). Whilst $U_{us}$ and power are more variable for the TLW situation (r7Sh-WMR, range 1.8 and 2.7 times greater, respectively, Table 3) than for the CNBL, the total power output is only 1.1 times lower

for the CNBL case (r3Nh-WMR) due to its higher $U_N$. Whilst these differences are small, if $U_N$ were equal for both cases, i.e., different far upstream windspeed, the TLW would cause more dramatic increases in power output compared to the control as the initial offset between the two cases would be removed. $U_{us}$ and power increases would be expected as the peak increases are not counter-balanced by the TLW troughs, as WMR is small and is not experiencing the lowest speeds in the TLW trough





in this situation (Fig. 16). Not accounting for differences in atmospheric stability and TLW impacts could result in over or
underestimation of power output when based on a mast measurement alone.

The CNBL case is approximated to real conditions at WMR so is more representative than purely neutral conditions,
however, what constitutes a true control for TLW situations is unclear. Here the CNBL has a shallow and weak inversion.
Modifications to height, depth and strength of inversion layers will produce different windspeeds and turbulence throughout
the domain and interact differently with individual turbines and whole wind farms. As discussed in section 5, investigating
TLWs using a variety of stability profiles, and producing control simulations with varying profiles and similar near upstream
windspeeds would be beneficial for full quantification of TLW impact.


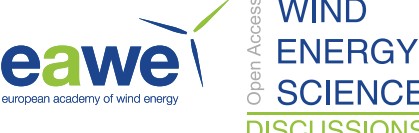

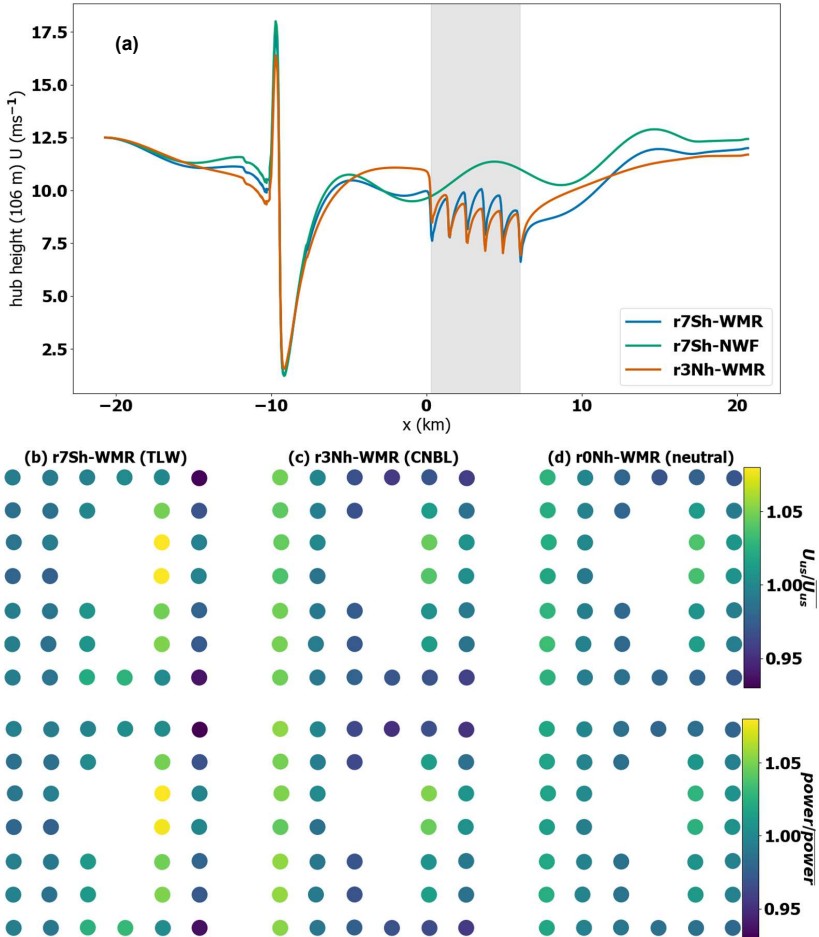

**Figure 16 a) hub height windspeed aligned with column 01 of WMR for TLW (r7Sh-WMR, blue line), TLW without WMR (r7sh-NWF, orange line) and weak CNBL case (r3Nh-WMR, green line). Below: Normalised $U_{us}$ and Power for WMR for TLW (column b, r7Sh-WMR), CNBL (column c, r3Nh-WMR) and neutral cases (column d, r0Nh-WMR).**




### 3.5 Impact of AGW on potential turbine loading for topographical AGW simulations

For all the topographical TLW simulations at WMR (r7Sh-WMR, r7Nh-WMR, x7Sh-WMR), the range of inflow angles is larger than for the neutral equivalents (r0Nh-WMR, x0Nh-WMR, Fig. 12). Mean inflow angle is largest for the TLW trough
simulation (x7Sh-WMR) where it is almost double that of the neutral equivalent (x0Nh-WMR) (mean 1.03°, 0.52°, range 0.98°, 0.14° respectively, Table 3). Whilst the TLW inflow values are higher, they are well within the tolerance range of modern wind turbines ($\leq \pm\, 8°$) so turbine fatigue loading does not seem to be a concern for these conditions. The TLW trough simulation (x7Sh-WMR) also shows the largest difference in turbine TI compared to the neutral case (r0Nh-WMR, mean 0.24, 0.16, range 0.11, 0.05 respectively). As the trends in turbulence and TI match, the changes in TI are likely a result of shear
associated with up and downslope TLW flow.

    Whilst this research focuses on the impact of TLWs on $U_{us}$ and power output, larger inflow angles, greater TI and associated shear suggests that some turbines across the wind farm are likely to experience greater fatigue loading during TLW events and that this is not uniform across the wind farm. However, these increases in inflow angle and TI do not appear large enough to substantially impact turbine fatigue and lifetimes.


### 4     Conclusions

    Topographical TLW interaction with wind farms is common and has, until recently, been overlooked. In this parametric study, turbine and whole wind farm windspeeds and power outputs behaved differently in the presence of topographically forced TLWs. In the simulations, the reference windspeed at the inlet was analogous to a mast measurement taken 20 km
upstream of a proposed wind farm site. In the presence of an upper layer inversion, strong TLWs meant the topographical influence was more apparent. These results demonstrate that with the same apparent synoptic forcing conditions, local conditions favouring TLW formation may lead to large deviations between the predicted and actual wind speed. Thus, power output from individual turbines and whole farms will vary significantly from predicted if these conditions are not accounted for. Greater variability in local turbulence and shear was also apparent during TLW situations attributed to TLW and capping
inversion impacts on wake and shear. However, the TLW impact on inflow angles within WMR, were well within the tolerance of modern wind turbines. TLW events affecting multiple windfarms may have less impact on power output for windfarms downstream of an existing windfarm due to appreciable reductions in TLW amplitude with windfarm interaction.

    The different atmospheric stability conditions led to the same upstream flow conditions interacting very differently with the topography upstream of the wind farms. Consequently, windfarm inflow speeds were highly variable between TLW and
non-TLW events leading to differences in windspeed throughout the windfarm. These differences were further complicated by the varying windspeed recovery from the coastal transition and differences in wind farm blockage effects in different stability regimes. Additionally, wake recovery appeared dependent on both the TLWs and strength of the capping inversions.



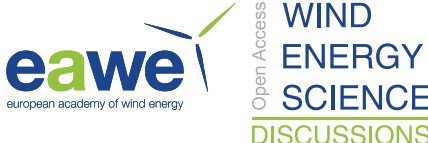

With these interacting conditions it was not possible to fully decouple which impacts on wind turbine and whole wind farm windspeeds and power were a result of TLWs and which were a result of different stability impacts on the flow.

When compared to purely neutral conditions throughout the domain, all the TLW simulations had reduced power output. These reduced speeds compared to neutral for TLW simulations were primarily due to reduced flow recovery after the hill due to stability differences. As it was not possible to define consistent wind farm inflow conditions between TLW and control simulations, it remains unclear how much of this influence was due to TLWs compared to impact of differing $U_N$. The effect will be situation dependent as differences in $U_N$ lead to different operating points on the thrust curve and non-linear changes

in wake losses. Yet, when compared to a non-TLW CNBL event at WMR, with higher $U_N$ than the TLW, subtle increases in turbine $U_{us}$ and turbine – whole wind farm power output were observed during the TLW. This suggests that TLWs may sometimes have beneficial impacts compared to real CNBL conditions. How the TLW impact is interpreted is largely based on what is taken as the 'control' situation. In all simulated cases, TLW events increased the variability in windspeeds and power outputs through the windfarm. Despite the variation in $U_N$ which complicates the interpretation of the results, it is

concluded that TLWs can have a substantial impact on the variation in wind speeds and TI experienced across an offshore wind farm and the resulting power output of individual wind turbines.

The location of the wind farm in the wave cycle was an important factor in determining the magnitude of TLW impacts. TLW peaks countered wake losses and TLW troughs enhanced them. There were greater ranges of windspeed and power output during TLW events; the range was greater for TLW trough than peak cases at WMR. Trough windspeeds were, however,

coincident with operating points on the power and thrust curves where wake losses were greater. Whether TLW impacts are beneficial, detrimental or balance out will be dependent on: the windfarm location within the TLW wave cycle, windfarm dimensions relative to the TLW wavelength and amplitude, TLW wavelength, TLW amplitude, and TLW orientation in relation to the windfarm dimensions. Whether the wave is quasi-stationary or travelling will also have an impact. A travelling TLW will have transient impacts on turbine outputs that may cancel out overall, whilst quasi-stationary waves may lead to

longer term differences compared to predicted power output. Again, the interpretation of TLW impact for all wind farm sizes will largely be determined by what reference conditions the TLW conditions are compared to. For example, when compared to purely neutral conditions, (not existing in reality), TLWs may lead to power output improvements compared to real atmospheric non-TLW situations for the same value of $U_N$. Without a constant $U_N$ between simulations it was not possible to determine whether there was a balancing effect across the wind farm. Furthermore, it is not yet known whether multiple TLW

events at the same windfarm may balance out over a longer period.

## 5 Future work recommendations

In the current work, it remains unclear how much contribution the following conditions make in TLW situations: (i) Differences in initial $U_N$, (ii) windspeed recovery from topographical obstacles, (iii) flow adaptation to roughness and temperature changes





after the land-sea transition, (iv) wind farm flow blockage, (v) TLW phase, (vi) height and strength of inversion layer, (vii) presence of surface and/or upper-level stability. Further work to obtain consistent $U_N$ would help quantify influences of the other variables listed above. Whilst subtle differences were found between the stable and neutral surface layer TLW conditions in the current work, applying a variety of surface stability conditions would provide a clearer understanding of the interaction between TLWs and the surface conditions. Additionally, varying the inlet wind speed and direction, inversion strength, depth

and height, topography dimensions and orientation would help determine their contributions to TLW impacts.

In the current work high resolution SCADA data was not available for demonstrating the impact of TLW in a real operational windfarm. Thus, this is a priority for future TLW investigations. For a fuller description of real atmospheric TLW-windfarm interactions moving forward, combined use of CFD, LiDAR, high temporal resolution SCADA and high-resolution mesoscale modelling to downscale ERA5 data is recommended. These methods would enable improved spatial and temporal

description of TLW characteristics which could then be utilised to assess the impact of TLWs on wind farms. Assessing different TLWs would provide information on the dependence of impacts on the TLW characteristics. Now theoretical TLW-windfarm impacts have been demonstrated, developing models for larger wind farms and existing wind farm clusters will demonstrate the impact of greater spatial interaction with TLWs. Whilst it may be possible to model a longer domain length in CFD, a coupled micro-mesoscale model would be more appropriate for this large problem. With larger windfarms, a stronger

influence on TLW amplitude is expected, which may enhance or reduce the windspeed fluctuations for downstream windfarms.

**Code and Data availability**

The measurement and reanalysis data used in this paper are open source. ERA5 reanalysis data (ERA5, 2020) are available at https://cds.climate.copernicus.eu/cdsapp#!/home. SAR 10m windfield data are managed by DTU and available at

https://science.globalwindatlas.info/. The ANSYS Windmodeller simulations and data are available from the corresponding author on request.

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

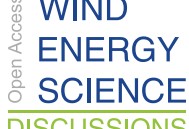

**Appendix A**

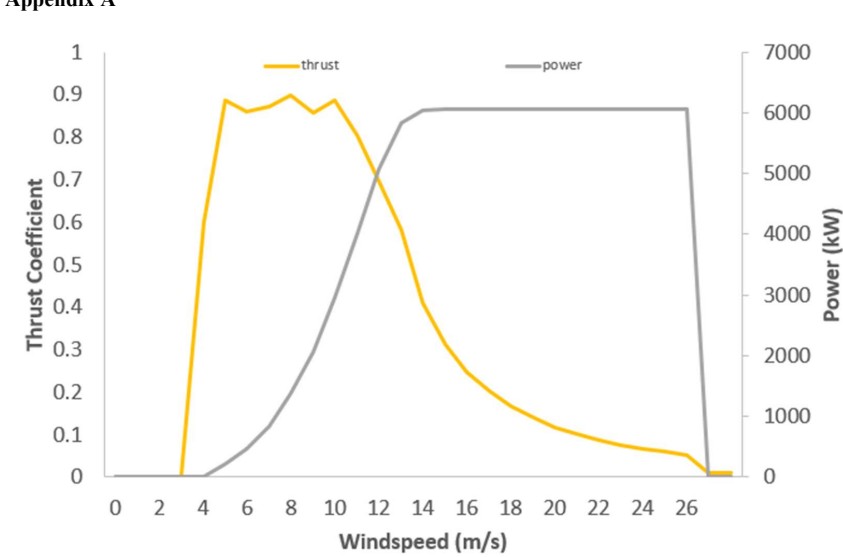

**Figure A. 1. Thrust coefficient and power curve data used for the 6MW 154m diameter turbines in the simulations**



## Appendix B

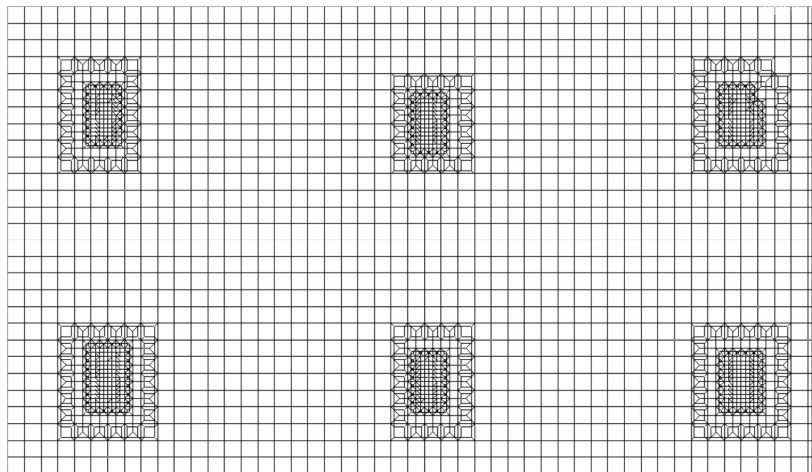

**Figure B. 1. Horizontal mesh structure for the WMR wind farm region of simulation domain at turbine hub height (106 m) used for all simulations. The mesh refinement around each actuator disk is shown in the darker regions. Note that mesh refinement around individual turbines leads to asymmetrical meshes for some turbines.**

## Appendix C

Fig. C. 1 shows the shear exponent factor ($\alpha$) for the neutral case (r0Nh-WMR) and the TLW case with a neutral surface layer (r7Nh-WMR). Shear is more variable within the TLW case where there are deeper near-wake losses (Fig. C. 2, Fig. C. 3). The greatest shear variability is experienced by F01 and F07 which experience the deepest near-wake losses and TI (Fig. C. 3) due to having a full column of turbines upstream. For the TLW case the wakes and elevated wake TI persist further downstream into the wave damping region. The impact of the TLW and the capping inversion are coupled so it is unclear which has a greater influence on the shear, turbulence and near wake loss depth is unclear.

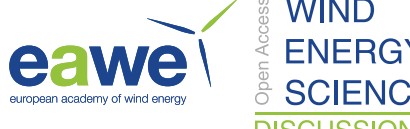

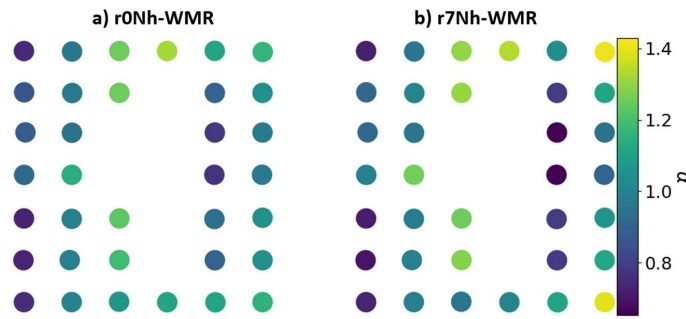


**Figure C. 1. WMR view from above - shear exponent factor (α) for a) r7Nh-WMR and b) r0Nh-WMR.**

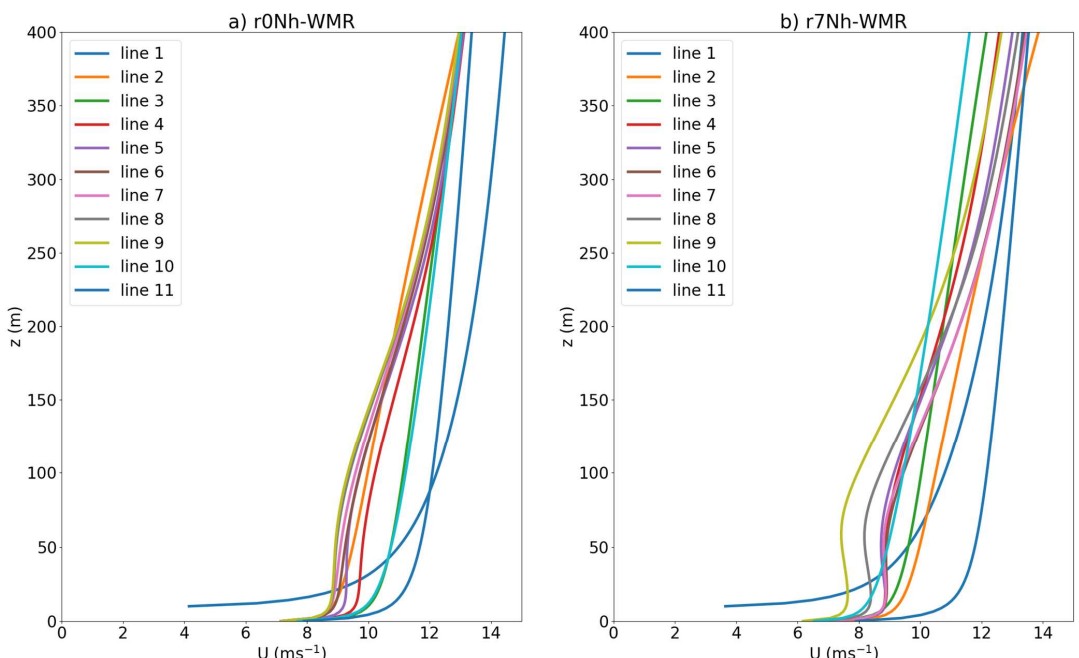

**Figure C. 2. Vertical windspeed profiles for a) r0Nh-WMR and b) r7Nh-WMR. Numbered lines correspond to lines 1-11 labelled in**
**Fig. 13a.**



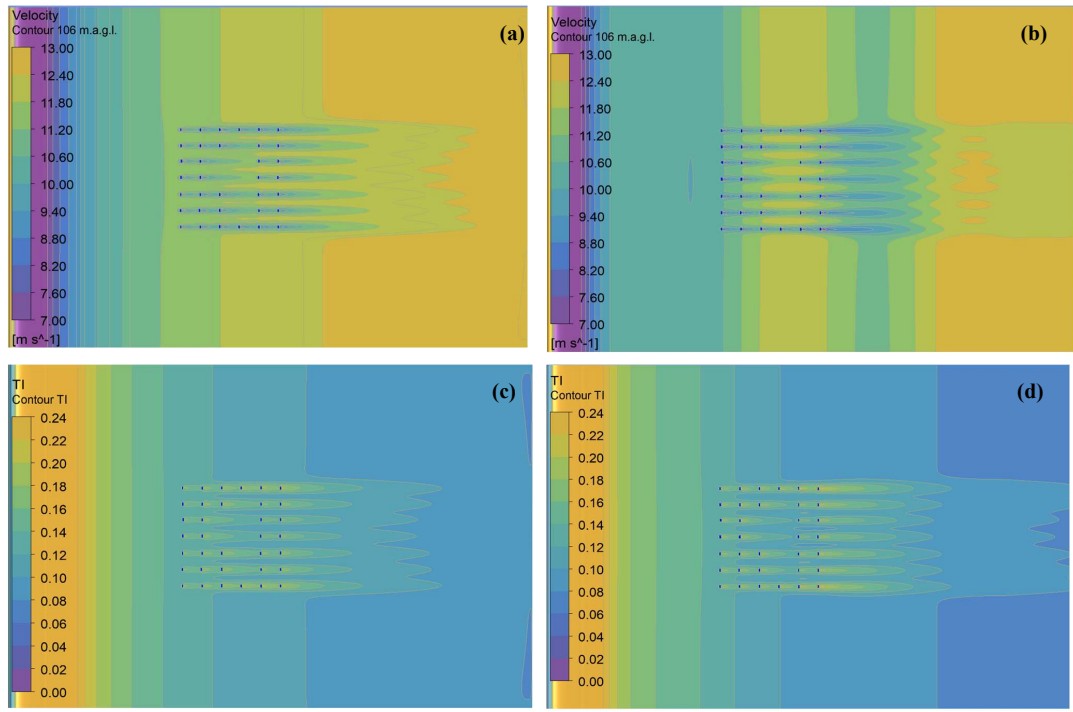

**Figure C. 3. WMR view from above in the CFD domain with colour contours of 106 m hub height windspeeds and TI for a), c) r0Nh-WMR and b), d) r 7Nh-WMR, see legend for windspeeds.**

### Acknowledgments

This research was funded by a NERC-CASE PhD Studentship with the ORE Catapult and undertaken at Loughborough University.