# Peer review of "Modelling the impact of trapped lee waves on offshore wind farm power output"

_Wind Energy Science, 2022_

## Author Comment (AC1)

We thank the reviewers for taking the time to provide helpful comments and suggestions. Your input has helped us improve this paper. We address your specific comments below.

RC1 response

1. (TLWs) as a "special case" (or one of possible cases) of AGWs. Therefore, "Atmospheric Gravity Waves, or Trapped Lee Waves (TLWs) (line 8)"……..might need to be a bit modified (AGWs are not always TLWs, but TLWs are always AGWs)……. should be consistent throughout the manuscript

*Thank you for drawing our attention to this error, text updated to clarify that TLWs are a special case of AGWs - line 8 and references to AGWs in section 3.3 onwards amended to refer to TLWs.*

2. Also, looking at the atmospheric gravity waves as waves that will appear if there is disturbance in vertical direction, how coastal transition fits in this description? Authors marked coastal transition in Fig. 9, but (maybe because of the scale of the plot) it is not visible if it affects the flow. If this transition is significant, that should be included somewhere in the text.

*The coastal transition impact is superimposed onto the flow recovery from the hill so the coastal transition has a subtle effect on the flow in this case and does not appear significant. It is labelled on the plot to explain the slight change in flow and show where the flow recovery from land begins.*

3. Line 103: Authors say they decided to go with much taller domain to avoid non-physical reflection of the gravity waves, but at the same time applied dumping layer on the top of their domain, to dump those same waves. So, is this choice of a taller domain sufficient to avoid the wave reflection, or there is still need for a dumping layer? Wouldn't be equally correct to use a bit shorter domain, with a dumping layer on the top? If not, why not.

*Preliminary trials without a damping layer at the top, regardless of domain height lead to wave reflection. A tall domain reduces wave reflection but damping is still required to dampen the waves. A tall domain which includes deep upper damping layer was employed in the current work after (Allaerts and Meyers, 2017) which also uses a 25 km high domain including 10km deep damping layer.*

4. Section 2.3: Authors referred to several studies that used dumping layer and how they setup their runs, but authors describe their choice of parameters as "trial and error". It would be useful if authors included a bit more details about their "trial and error" journey.Did they learn anything in that process? Anything worth of including in a manuscript?

*The authors used ideas from models in the literature as a starting point. They then modified domain dimensions, damping layer location, thickness, and depth to find a set-up that converged to a steady state when TLWs were present. This set up will be specific to the domain dimensions and contents (in this case a hill, sea surface, windfarm), distance between the objects within the domain, atmospheric conditions etc). The trial and error process was stopped when wave reflection was low enough to enable a steady state solution. Findings – inflow damping lead to wave reflection upstream of the hill, too shallow domain lead to wave reflection from the top and insufficient outflow damping lead to reflection at the outflow.*

*Text added to the end of section 2.3: "Key findings during this process were that increased domain length and depth with outflow and upper-level damping resolved most wave reflection. We did not find inflow damping helpful and the damping layers thinner than those used were insufficient. However, these settings are specific to the dimensions, contents and atmospheric conditions of the domain used in this research."*

5. Line 219: There is no Fig. 5a

*Corrected to (Fig. 5, Fig. 6).*

6. Fig 9: Is this time average? I'm guessing this is an evolution of the mean wind speed. That should be stated in the label. (Same comment applies for Fig. 11)

*This is windspeed at a constant height of 106m above the surface once the solution has reached a steady state with standing waves. Although the final solution is steady state, A URANS solver was used to ensure numerical stability.*

*Line 42 updated: "Thus, this research investigates the influence of TLWs on offshore wind farm power output using Unsteady Reynolds Averaged Navier Stokes (URANS) CFD simulations. Although this work focuses on resolving standing waves, a URANS solver was preferred for reasons of numerical stability."*

*Added to start of section 3.1: "The results from TLW simulations in this section are a snapshot from when the simulations reached a steady state with standing waves."*

7. Figure 10: I'm guessing this is a snapshot of a horizontal and vertical wind speed after the field reached a steady state. I think that it would be helpful for a reader if Fig. 10a included x-coordinate (similar to Fig. 4). Also, Figure 10b should include z-coordinate.

*Yes this is a snapshot after reaching a steady state, this should be clearer with the above edit. Fig 10a, 4, 10b now have axes and labels for all co-ordinates.*

8. Line 319: "However, it is unclear at this stage whether the upstream peak is an artefact of imperfect wave damping and upstream domain length". This is a good example where playing with dumping layer parameters is useful and helps with making less "unclear" conclusions. (is that upstream behavior the same or is it different for different dumping layer values?)

*We agree that playing with damping layer parameters would be useful and have added this to future work recommendations (section 5).*

*"Additionally, future investigations into TLWs would benefit from systematic adjustment of wave damping and domain dimension parameters to develop guidelines for optimum wave damping set-up."*

*However, at this stage we believe the upstream peak is far enough from the windfarm that it would not have a significant effect on the solution. The main focus of this work was to assess potential impacts of topographical TLWs.*

9.  Line 320: "consistent with findings in previous studies' it would be useful for a reader to include a few words on what previous studies found

*The line above states "Notably, there is a TLW peak upstream of the hill in Fig. 10b; TLW peaks also occurred in mathematical models of wind-farm induced TLWs where the Froude number ($Fr$, Eq. **Error! Reference source not found.**) was less than 1 (Smith, 2010; Lanzilao and Meyers, 2021)."*

*__LINE 336__ updated to be clearer "In the TLW cases (r7Sh-WMR, r7Nh-WMR) $Fr \sim 0.04$, consistent with upstream wave occurrence when $Fr < 1$ in previous studies (Smith, 2010; Lanzilao and Meyers, 2021)"*

10. Line 346: I think authors should include here some sort of an introductory sentence for next section and Fig. 12. It is a bit confusing to just jump into analysis related to Fig 12, without any previous introduction that following analysis is looking at the flow within the windfarm itself.

*Updated to include "Here we introduce TLW impacts at WMR at the windfarm level by reviewing individual turbine and whole windfarm flow."*

11. *Looking at Figure 11 (11a and 11b), it is interesting to notice that two runs (x7Sh-WMR and s7Sh-WMR) have a bit different wind speed patterns at 106m, even though the only difference between two runs is the length of a domain (the only difference between two runs in a domain size in x-direction, right?)*

*Yes, the only difference is the domain length. As the windspeed profile is set at the inlet with a Zilitinkevitch profile it is not trivial to ensure both simulations have identical flow evolution throughout the domains with different distances between the inlet and windfarm. To mitigate this upstream divergence a significantly longer upstream domain length would be required and consequently wave damping settings would need to be adjusted. A substantial amount of computing resource would be required to run and tune these simulations. Whilst this is recommended for future work, at this stage the main focus of the work was to demonstrate that TLWs can impact windfarm flow rather than design an optimum TLW model.*

*The discrepancy in wind speed is considered small enough and far enough upstream from the main area of interest (the windfarm) to have a negligible impact on the solution at the turbines.*

*Added to section 5 future work: "Further work to determine the relative contributions to wake recovery by the stable surface layer and the capping inversion aloft would need to first address the evolution of the stable surface layer from the inlet to the windfarm. This may be achieved by a considerably longer upstream domain length and exaggerated upstream surface stability."*

12. Figure 13: line 1 and line 11 look like they are the same color. Maybe to change color of line 1 to black. (same applies to Fig. C.2. Also, this figure is missing x-axis. It would be nice to see if for example line 2 is in offshore environment or not.
*Line 1 has been changed to black in both plots. X axis and labels added.*

13. Overall, with a few little changes and edits, I think this manuscript is making a significant contribution to a current body of literature looking at the impact of atmospheric gravity waves on wind farms situated in offshore environment.
*Thank you for your contributions.*
* * *
This paper discusses the impact of trapped lee waves on offshore wind farm by conducting several idealized CFD simulations. The topic is certainly relevant to the wind energy community. It is very interesting to see that the location of wind farm in the TLW cycle can have such a significant impact on the simulated power output. Overall, I think the paper can be accepted for publication after addressing some minor comments.

1. I think authors should better summarize section 2.3 as it is too lengthy. A lot of texts (Lines 130 – 160) are literature review which should be include in the introduction section but not here.

Section 2.3 has been shortened and the literature review element has been moved to a new section 1.1 Wave Damping

2. Please label the x and y axis in Figure 10a and 10b.
Labelled as suggested.

3. Line 312: "Notably, there is a TLW peak upstream of the hill in Fig. 10b", can you label that in the plot?
Labelled as suggested.

4. I think the label in Figure 11 is wrong, it should be (r7Sh-NWF, green line) and (r0NHWMR, orange line).

Corrected as suggested.

5. "The greater recovery is explained by the increases in windspeeds due to the TLW countering wake losses. However, the central gap in WMR makes the TLW effect less clear.". This is actually very interesting. It seems like wind speed over the wind farm recovers faster over stable surface condition than that over neutral. Is it possible for authors to run a test case with surface stability in between stable and neutral to see how the wind speed recovers. Or fill in the gap with wind turbines and see whether there is still such significant wind speed recovery? I am wondering whether that is just some artificial noise from the numerical solution.

*There was an error in the plotting. The corrected plot below shows that the windspeed recovers faster after the central gap when the TLW is reaching its peak at row E, whilst the recovery is slower for neutral and trough cases. There is little difference between the stable surface layer and the neutral surface layer in the peak cases (r7Sh-WMR and r7Nh-WMR, respectively). We used this layout as it was an existing operational windfarm layout and some modern windfarm layouts also include a central gap.*

[Figure]

[Figure]

*It would be interesting to see the impacts on different wind farm layouts. Added to section 5 future work: "It is recommended that future work investigations use a variety of windfarm layouts to investigate wake recovery under TLW conditions."*

6.  Lines 425, "As the TLW persists with reduced amplitude after interaction with WMR, this suggests that a TLW event affecting multiple farms may have less impact on windspeed and power fluctuations if there is another windfarm upstream." Can the authors test this by setting up two artificial wind farms in their simulation domain?

*The impact on downstream windfarms is interesting and has been highlighted in future work section "Now theoretical TLW-windfarm impacts have been demonstrated, developing models for larger wind farms and existing wind farm clusters will demonstrate the impact of greater spatial interaction with TLWs. Whilst it may be possible to model a longer domain length in CFD, a coupled micro-mesoscale model would be more appropriate for this large problem."*

*It may be beyond the limits of a RANS model to accurately represent this scenario. To test this a large amount of computational resource would be needed to extend the domain and tune the wave damping settings to introduce a secondary downstream windfarm in a physically realistic way. The authors believe it is more a priority to find the optimum settings for wave damping in future work before the models are ready to be tested in this way. As mentioned in section 5, another modelling methodology may be more suitable for this question.*

7.  Line 455: As the surface stability temperature offset reduces substantially after interaction with the topography and sea surface (lines 2-11, Fig. 13), the stable layer is relatively shallow with the surface lapse rate increasing to near neutral conditions around rotor height. Thus, the differences between r7Sh-WMR and r7Nh-WMR are relatively subtle. Is it possible to have a fixed lapse rate in the simulation so that we can see the impact of stability more clearly?

*Ideally a fixed lapse rate would be preferred. As the windspeed profile is set at the inlet with a Zilitinkevitch profile it is not trivial to ensure the temperature profile is fixed throughout. Whilst we think this is a point to be addressed in future work, we believe the main focus of the work (TLW impacts on offshore wind farms) is addressed without these modifications.*

*Added to section 5: "Further work to determine the relative contributions to wake recovery by the stable surface layer and the capping inversion aloft would need to first address the evolution of the stable surface layer from the inlet to the windfarm. This may be achieved by a considerably longer upstream domain length and exaggerated upstream surface stability."*